# EXAGREE: TOWARDS EXPLANATION AGREEMENT IN EXPLAINABLE MACHINE LEARNING

## ABSTRACT

Explanations in machine learning are critical for trust, transparency, and fairness. Yet, complex disagreements among these explanations limit the reliability and applicability of machine learning models, especially in high-stakes environments. We formalize four fundamental ranking-based explanation disagreement problems and introduce a novel framework, EXplanation AGREEment (EXAGREE), to bridge diverse interpretations in explainable machine learning, particularly from stakeholder-centered perspectives. Our approach leverages a Rashomon set for attribution predictions and then optimizes within this set to identify Stakeholder-Aligned Explanation Models (SAEMs) that minimize disagreement with diverse stakeholder needs while maintaining predictive performance. Rigorous empirical analysis on synthetic and real-world datasets demonstrates that EXAGREE reduces explanation disagreement and improves fairness across subgroups in various domains. EXAGREE not only provides researchers with a new direction for studying explanation disagreement problems but also offers data scientists a tool for making better-informed decisions in practical applications.

## 1 INTRODUCTION

As machine learning models gain prominence in critical fields such as healthcare, science and finance, the demand for transparent explanations of their predictions has intensified, particularly in high-stakes decision-making scenarios (Kailkhura et al., 2019; Wiens & Shenoy, 2018; Carvalho et al., 2022; Agarwal et al., 2022; Ghassemi et al., 2021). However, a significant challenge has emerged: explanation disagreement, where explanations from different methods or models conflict with each other (Krishna et al., 2022; Rudin, 2019; Li & Barnard, 2024). This disagreement hinders the potential impact and trustworthiness of machine learning models, especially when the consequences of model decisions can have significant real-world impacts.

Explanation disagreement stems from multiple, complex sources. Multiple model-agnostic post-hoc explanation methods often yield inconsistent results for the same model and prediction (Krishna et al., 2022). The involvement of various stakeholders, each with unique expertise and objectives, further complicates model explanation (Imrie et al., 2023; Binns, 2018; Hong et al., 2020). Even

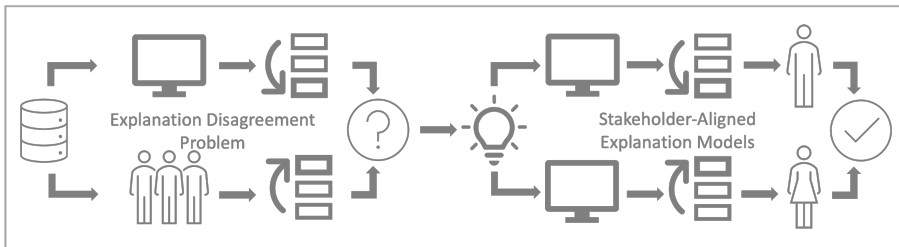

Figure 1: Addressing the explanation disagreement problem with EXAGREE. Left: Illustration of explanation disagreement, where machine learning model explanations conflict with stakeholders' requirements, needs, or aims. Right: EXAGREE's solution - identifying an SAEM from a Rashomon set that maximizes agreements with diverse stakeholder expectations.

interpretable models, which are theoretically more transparent, can produce explanations that diverge from stakeholder expectations or domain knowledge. Moreover, the existence of multiple well-performing models for a given task, known as the Rashomon set, introduces another layer of variability in explanations (Fisher et al., 2019; Rudin, 2019; Dong & Rudin, 2020; Ghorbani et al., 2019; Adebayo et al., 2018). Traditional approaches have primarily focused on developing new explanation methods or improving model interpretability. However, limited attention has been given to addressing the fundamental issue of explanation disagreement (Krishna et al., 2022; Li & Barnard, 2024). More related works are provided in the Appendix B.

To bridge this gap, we introduce EXplanation AGREEment (EXAGREE), a novel framework designed to enhance explanation agreement in explainable machine learning under ranking supervision. EXAGREE adopts a stakeholder-centered approach that prioritizes the satisfaction of diverse human needs, leveraging the Rashomon set concept to identify Stakeholder-Aligned Explanation Models (SAEMs) that provide more fair, faithful, and trustworthy explanations, as illustrated in Fig. 1 (Fisher et al., 2019; Hsu & Calmon, 2022; Dong & Rudin, 2020; Li & Barnard, 2023b; Rudin et al., 2024).

Our work makes several significant contributions to the field of explainable machine learning:

- We formalize four fundamental explanation disagreement problems in Sec. 2: stakeholder disagreement, model disagreement, explanation method disagreement, and ground truth disagreement, providing a structured foundation for future research in the field.

- By emphasizing the purpose of explanations, we reframe these complex challenges from a stakeholder-centered perspective in Sec. 2, aiming to satisfy diverse human needs. This novel viewpoint offers a potential pathway for resolving these disagreement conflicts.

- We propose EXAGREE, the first framework that aims to enhance both explanation faithfulness and fairness by leveraging the Rashomon set concept, mitigating explanation disagreement while preserving model performance in Sec. 3. This approach utilizes the potential of model disagreement as a powerful means to resolve stakeholder disagreement.

- Through rigorous empirical analysis in Sec. 4 on the OpenXAI (Agarwal et al., 2022) disagreement measurement benchmark, we gain new insights into the nature of explanation disagreement. Our experiments demonstrate EXAGREE's effectiveness in identifying SAEMs that improve explanation agreement for diverse stakeholders.

## 2 PRELIMINARIES

**Problem Statement** We formalize the explanation disagreement problem in the context of feature attribution for machine learning models, where the attribution assigned to a feature is a measure of that feature's importance to the model's prediction (Krishna et al., 2022; Sundararajan & Najmi, 2020). We denote a set of good models, feature attributions, and their corresponding rankings in this framework, with a summary of notations provided in Appendix F Table 6.

Let $\mathcal{M}$ be a set of good models (a Rashomon set), where each model in the set meets a predefined performance threshold $\epsilon$ for a given task $(\mathbf{X}, \mathbf{y}) \in \mathbb{R}^{n \times (p+1)}$, as defined by Fisher et al. (2019), Dong & Rudin (2020), Xin et al. (2022), and Li & Barnard (2023a):

$$\mathcal{M} = \{M : \mathcal{L}(M(\mathbf{X}), \mathbf{y}) \leq (1 + \epsilon)\mathcal{L}(M^*(\mathbf{X}), \mathbf{y})\}. \tag{1}$$

For a model $M \in \mathcal{M}$ and an explanation method $\varphi \in \Phi$, we calculate feature attributions as: $\mathbf{a}^{M,\varphi} = (a_1^{M,\varphi}, a_2^{M,\varphi}, \ldots, a_p^{M,\varphi})$. These attributions yield a ranking: $\mathbf{r}^{M,\varphi} = (r_1^{M,\varphi}, r_2^{M,\varphi}, \ldots, r_p^{M,\varphi})$, where $r_i^{M,\varphi}$ represents the rank of feature $i$. The ranking can be derived based on the ordering of attributions: $a_{(1)} \succ a_{(2)} \succ \ldots \succ a_{(p)}$, where $a_{(i)}$ denotes the $i$-th largest attribution. For interpretable models $M_{\mathcal{I}} \in \mathcal{M}_{\mathcal{I}} \subset \mathcal{M}$, such as decision trees and linear regressors, we can obtain ground truth attributions $\mathbf{a}_{\text{true}}^{M_{\mathcal{I}}}$ and rankings $\mathbf{r}_{\text{true}}^{M_{\mathcal{I}}}$. Feature attributions and rankings serve as explanations for model behaviour and form the basis for understanding how different explanations may conflict with each other.

### 2.1 RANKING-BASED EXPLANATION DISAGREEMENT PROBLEM

While the broader issue of explanation disagreement has been discussed in existing literature, the focus has been fragmented. For instance, Rudin (2019) focused on model-related issues, Sundarara-

jan & Najmi (2020) examined explanation method variations, Miller (2023) explored stakeholder perspectives, and Krishna et al. (2022) investigated the disagreement problem from a practitioner's viewpoint. We extend beyond these individual focuses by explicitly formulating a ranking-based disagreement problem across multiple scenarios:

1. **Stakeholder Disagreement:** Different stakeholders in $\mathcal{S}$ may prefer different rankings

$$\exists k, l \in \mathcal{S}, k \neq l : \mathbf{r}^k \neq \mathbf{r}^l.$$

   This disagreement can arise from varying expertise, requirements, or objectives among stakeholders. For instance, a data scientist might prioritize statistical significance, while a domain expert may value features based on domain knowledge. These divergent perspectives can lead to conflicting interpretations of model behaviour and decision-making processes.

2. **Model Disagreement:** Different models in $\mathcal{M}$ can produce different rankings

$$\exists M, M' \in \mathcal{M}, M \neq M', \varphi \in \Phi : \mathbf{r}^{M,\varphi} \neq \mathbf{r}^{M',\varphi}.$$

   This scenario occurs when multiple models with similar performance yield different feature attributions, considering a Rashomon set. For example, a linear regression model and a neural network might assign different importances to features, despite achieving comparable predictive accuracy.

3. **Explanation Method Disagreement:** Different explanation methods in $\Phi$ can yield different rankings for the same model

$$\exists M \in \mathcal{M}, \varphi, \varphi' \in \Phi, \varphi \neq \varphi' : \mathbf{r}^{M,\varphi} \neq \mathbf{r}^{M,\varphi'}.$$

   This scenario highlights the variability in post-hoc explanation methods. For instance, LIME and SHAP, two popular explanation methods, might provide different feature importance rankings for the same model.

4. **Ground Truth Disagreement:** Ground truth interpretations from interpretable models can conflict with post-hoc explanations or stakeholders' needs

$$\exists M_\mathcal{I} \in \mathcal{M}_\mathcal{I}, \varphi \in \Phi : \mathbf{r}^{M_\mathcal{I},\varphi} \neq \mathbf{r}^{M_\mathcal{I}}_{\text{true}}; \exists k \in \mathcal{S}, M_\mathcal{I} \in \mathcal{M}_\mathcal{I} : \mathbf{r}^k \neq \mathbf{r}^{M_\mathcal{I}}_{\text{true}}.$$

   For example, a post-hoc explanation method applied to a linear regression model might yield feature importances that differ from the model's coefficients.

**Remark 1.** *While these scenarios provide a foundation for understanding explanation disagreements, practical situations often involve more complex, interconnected challenges, named **compound disagreements**. Different models may respond differently to various explanation methods ($\exists M, M' \in \mathcal{M}, \varphi, \varphi' \in \Phi : \mathbf{r}^{M,\varphi} \neq \mathbf{r}^{M,\varphi'} \neq \mathbf{r}^{M',\varphi} \neq \mathbf{r}^{M',\varphi'}$). In more complex cases, disagreements may span stakeholders, models, and methods.*

## 2.2 PRIMARY OBJECTIVE

To address the explanation disagreement problem, we aim to **identify a well-performing model that minimizes disagreement (or maximizes agreement) between model explanations and stakeholder expectations.** This approach is rooted in the principle of human-centered decision-making (Aggarwal et al., 2016; Miller, 2019), which prioritizes satisfying the requirements of relevant stakeholders.

By focusing on the specific needs of stakeholders, we intentionally simplify the complex nature of compound disagreements, allowing for a more targeted and practical solution. We formalize this as an end-to-end optimization problem, considering each stakeholder's expectations independently while maintaining model performance:

$$\min_M \quad \mathcal{O}_i(\mathbf{r}^{M,\varphi}, \mathbf{r}^i) \quad \forall i \in \mathcal{S}, \quad \text{s.t.} \quad \mathcal{L}(M(\mathbf{X}), \mathbf{y}) \leq \tau \tag{2}$$

where $\mathcal{O}(\cdot, \cdot)$ is a suitable disagreement measure between rankings, $\tau$ is the performance threshold, and $\mathbf{r}^i$ is the expected ranking of stakeholder $i$. By treating each stakeholder $i \in \mathcal{S}$ separately, we address their unique expectations, using their individual feature rankings as distinct optimization targets. In this work, we will focus on exploring a set of well-performing models to achieve this objective, detailed in Sec. 3.

**Remark 2.** *This objective is general and open-ended due to the nature of the explanation problem. The choice of explanation method and disagreement metrics are left unspecified here, allowing for further exploration and development in subsequent studies.*

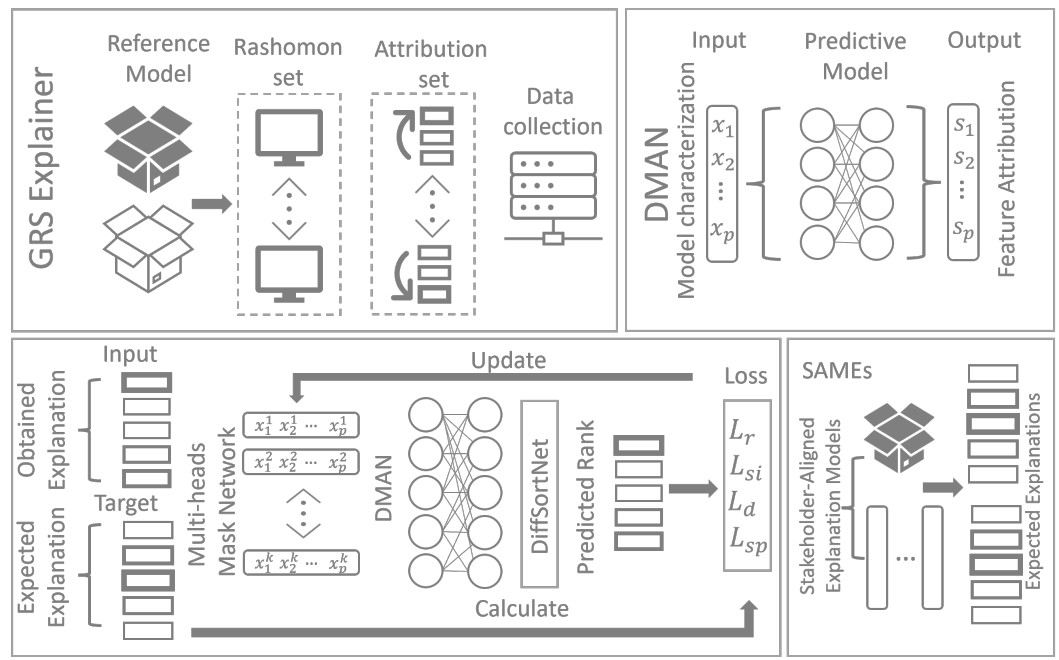

Figure 2: The EXAGREE framework overview, illustrates the two-stage process of EXAGREE from top-left to bottom-right. First stage: *Rashomon Set Sampling and Attribution Mapping (top)* approximates the Rashomon set and generates the attribution set $\mathcal{D}_{att}$. $f_{dman}$ is trained on $\mathcal{D}_{att}$ to map model characterizations to feature attributions. Second stage: *SAEM Identification (bottom)* aims to identify SAEMs under stakeholder's target ranking supervision, incorporating $f_{dman}$ and $f_{diffsort}$. The entire process is designed to be differentiable for end-to-end optimization.

## 2.3 EVALUATION MATRICES

To comprehensively assess the disagreement between explanations, we employ a set of quantitative metrics adapted from the OpenXAI benchmark (Agarwal et al., 2022). These metrics evaluate both the faithfulness and fairness of explanation methods. The core principle underlying our evaluation is that higher agreement between rankings implies greater faithfulness, which can be formalized as:

$$\text{agreement} \propto \text{faithfulness} \propto -\mathcal{O}(\mathbf{r}, \mathbf{r}^*)$$

Our evaluation includes *faithfulness assessment* and *fairness assessment*. Faithfulness measures how accurately an explanation reflects the true behavior of the underlying model, while fairness assesses the consistency of explanation quality across different subgroups (Dai et al., 2022). Faithfulness metrics contain Feature Agreement (FA), Rank Agreement (RA), Sign Agreement (SA), Signed Rank Agreement (SRA), Pairwise Rank Agreement (PRA), Rank Correlation (RC), Prediction Gap on Important feature perturbation (PGI), and Unimportant feature perturbation (PGU). Detailed descriptions are presented in the Appendix Table 3.

While our study primarily focuses on global-level feature attributions, we adapt these metrics, originally designed for local-level attributions, by averaging feature attributions across all instances to obtain global attributions. For a detailed description of each metric, please refer to the Appendix.

**Remark 3.** *An ideal explanation should perform well across all assessment metrics. However, explanations may excel in one metric while performing poorly in another, making it challenging to evaluate the overall agreement. To address this in practice, we count the number of times each explanation method achieves the best value across the entire set of metrics.*

## 3 EXAGREE: EXPLANATION AGREEMENT FRAMEWORK

EXAGREE is designed to mitigate the challenge of explanation disagreement, by employing a two-stage workflow, as illustrated in Fig.2:

1. *Rashomon Set Sampling and Attribution Mapping*: This preparatory stage involves two key steps: a) Approximation of similar-performing models from the given dataset using the General Rashomon Subset Sampling (GRS) algorithm (Li et al., 2024). b) Training a Differentiable Mask-based Model to Attribution Network (DMAN) that maps feature attributions from model characterizations for use in the next stage.

2. *Stakeholder-Aligned Explanation Model (SAEM) Identification*: In this stage, we identify explanation models that align with stakeholder requirements within the approximated Rashomon set. This is achieved by optimizing a Multi-heads Mask Network (MHMN), which: a) Incorporates the previously trained DMAN for feature attribution mapping. b) Incorporates a Differentiable Sorting Network (DiffSortNet Petersen et al. (2022)) to enable ranking supervision.

The following subsections provide detailed discussions of these components and their roles in the overall framework.

## 3.1 Stage 1: Rashomon Set Sampling and Attribution Mapping

The concept of Rashomon sets provides a powerful framework for addressing the challenge of explanation disagreement (Rudin et al., 2024), which allows us to transform single attribution values into ranges, and enables the search for models with stakeholder-expected rankings within a set of similarly performing models. Rashomon sets naturally relax the performance constraint in our optimization problem and mitigate ground truth disagreement by considering all candidate models within the Rashomon set as viable options. This allows us to reformulate our original optimization problem (Eq. (2)) into:

$$\min_{M \in \mathcal{M}} \quad \mathcal{O}_i(\mathbf{r}^{M,\varphi}, \mathbf{r}^i) \quad \forall i \in \mathcal{S}. \tag{3}$$

While various methods exist to explore Rashomon sets Hsu & Calmon (2022); Dong & Rudin (2020); Fisher et al. (2019), we adopt a general Rashomon set sampling algorithm for its generalizability and implementation sparsity to meet our practical requirements (Li et al., 2024). This approach: a) approximates the Rashomon set for a reference black-box model $M$, assuming stakeholders are given a black-box model with unknown explanations. b) guarantees a fair and consistent comparison of explanations by using a model-agnostic approach, thus relaxing explanation method disagreement.

Our goal is to identify specific models within this set that meet the desired ranking based on calculated attributions. However, optimizing a model under ranking supervision requires a differentiable mapping function from a model to the corresponding feature attributions, which presents two main challenges: a) model representations in the set as further input. b) the non-linear relationship between feature attributions and model representations.

**Remark 4.** *It's important to note that there is **no guarantee** that a model with an expected ranking can be found within a Rashomon set. A detailed discussion of ranking in the Rashomon set and proof of this concept is provided in the Appendix D.*

**Differentiable Mask-to-Attribution Network (DMAN)** The above challenges are addressed through: a) the fact that all models in the sampled set can be characterized by masks, providing uniform representations for different models in the Rashomon set. b) a differentiable mapping function from model characterizations (masks) to feature attributions.

We propose the DMAN $f_{dman}$ as a surrogate model that bridges the gap between models in the Rashomon set and their feature attributions. DMAN is a neural network trained to approximate the relationship between masks (representing models in the Rashomon set) and their corresponding attributions. The training process uses a dataset $\mathcal{D}_{att} = \{\mathbf{m}_{\mathcal{R}}, \mathbf{A}\}$, where $\mathbf{m}_{\mathcal{R}}$ are masks and $\mathbf{A}$ are corresponding attributions. The parameter optimization of the network is expressed as:

$$f^*_{dman,\theta} = \arg\min_{\theta \in \Theta} \mathcal{L}_{\text{MSE}}(f_{dman,\theta}, \mathcal{D}_{att}) \tag{4}$$

While DMAN provides an approximation as a surrogate model, its accuracy is crucial for the overall framework. To ensure reliability, we calculate actual attributions when evaluating the final results. This stage allows us to efficiently utilize the Rashomon set for attribution prediction while maintaining a differentiable pipeline for further optimization.

## 3.2 STAGE 2: SAEM IDENTIFICATION

Building on the prior stage, SAEMs can be searched within the approximated Rashomon set. This requires a differentiable function that maps feature attributions to target rankings.

**Ranking Supervision and Correlation Metric** We employ monotonic differentiable sorting networks in our framework, utilizing the cumulative density function (CDF) of the Cauchy distribution $f_{\mathcal{C}}$ from the work of Petersen et al. (2022). This network, denoted as $f_{diffsort}$, enables ranking supervision where the ground truth order of features is known while their absolute values remain unsupervised.

We adopt Spearman's rank correlation (negative w.r.t agreement) as our measure of disagreement distance (in Eq. (3)) due to its differentiability (Dodge, 2008; Petersen et al., 2022; Huang et al., 2022). The correlation for a specific ranking from stakeholder $i$ can be calculated as:

$$\rho^i(\mathbf{r}^{M,\varphi}, \mathbf{r}^i) = \frac{\mathrm{Cov}(\mathbf{r}^{M,\varphi}, \mathbf{r}^i)}{\mathrm{Std}(\mathbf{r}^{M,\varphi})\,\mathrm{Std}(\mathbf{r}^i)} = \frac{\mathrm{Cov}(f_{diffsort}(|\mathbf{a}^{M,\varphi}|), \mathbf{r}^i)}{\mathrm{Std}(\mathbf{r}^{M,\varphi})\,\mathrm{Std}(\mathbf{r}^i)}. \tag{5}$$

It's important to note that feature attributions have directions that do not necessarily represent their strength. To address this, we use the absolute value of attributions in the correlation calculation, ensuring both positive and negative importances are appropriately accounted for in the ranking. In practice, stakeholders may or may not require information about the direction of feature importance. To accommodate this variability, we incorporate a sign loss in our optimization process when applicable. Loss function details are discussed in the following section.

### 3.2.1 MULTI-HEADS ARCHITECTURE

The complexity of stakeholder needs and the uncertainty of finding a model perfectly matching stakeholder-expected rankings within the Rashomon set necessitate a multi-head architecture.

For a single stakeholder group, we are motivated by the following lemma:

**Lemma.** *For a given target ranking, there may exist multiple distinct rankings that have the same Spearman's rank correlation coefficient with the target ranking (proof see Appendix E).*

When considering multiple stakeholder groups, the multi-head architecture becomes critical. This is motivated by an important observation:

**Proposition.** *The increase in disagreement among stakeholders leads to greater opportunity to find a more faithful model (proof see Appendix E), shown as:*

$$\mathbb{P}(\exists M^* \in \mathcal{M}, \rho(\mathbf{r}^{M,\varphi}, \mathbf{r}^j) < \rho(\mathbf{r}^{M^*,\varphi}, \mathbf{r}^j)) \propto (1 - \rho(\mathbf{r}^i, \mathbf{r}^j)), \tag{6}$$

*where $1 - \rho(\mathbf{r}^i, \mathbf{r}^j)$ is the disagreement between two stakeholders $i$ and $j$ and $i \neq j$.*

Consequently, we integrate multiple heads into the architecture, each corresponding to a potential solution. By integrating the above components $f_{dman}$ and $f_{diffsort}$ into the architecture, our objective function is reformulated to minimize negative ranking correlation across all heads in an MHMN:

$$\min_{\Theta} \mathcal{L}_{rank} = \min_{\Theta} \sum_{j=1}^{h} \min_{M_j \in \mathcal{M}} -\rho(\mathbf{r}^{M_j,\varphi}, \mathbf{r}^*),$$

where $\Theta$ represents the set of parameters for all $h$ heads and $\mathbf{r}^*$ is the target ranking.

### 3.2.2 ATTRIBUTION DIRECTION, SPARSE AND DIVERSE CONSTRAINTS

To ensure that our multi-head architecture produces meaningful and diverse solutions while respecting stakeholder input, we introduce several key constraints:

**Attribution Direction** ($\mathcal{L}_{sign}$): We recognize the importance of maintaining the direction of feature attributions as specified by stakeholders. To achieve this, we incorporate a sign loss:

$$\mathcal{L}_{sign} = \mathcal{L}_{MSE}(\mathrm{sign}(\mathbf{a}^{M,\varphi}), \mathrm{sign}(\mathbf{a}^*_{true}))$$

This loss term ensures that the sign of the attributions in our identified models aligns with the stakeholder-specified directions, when $\mathbf{a}^*_{true}$ provided as ground truth attributions. The corresponding target ranking is derived as $\mathbf{r}^*_{true} = f_{diffsort}(|\mathbf{a}^*_{true}|)$.

**Sparsity Constraint** ($\mathcal{L}_{sparsity}$) and **Diversity Constraint** ($\mathcal{L}_{diversity}$): To encourage both variation across masks and within each mask, we implement sparsity and diversity losses. These constraints aid in uncovering diverse explanations that are consistent with stakeholder expectations while providing a range of potential interpretations.

The overall objective for a stakeholder group, incorporating these constraints, is formulated as:

$$\min_{\Theta}(\mathcal{L}_{rank} + \mathcal{L}_{sign} + \lambda_1 \mathcal{L}_{sparsity} + \lambda_2 \mathcal{L}_{diversity}), \tag{7}$$

where $\lambda_1$ and $\lambda_2$ are hyperparameters that control the weight of the sparsity and diversity losses. By propagating the error backward through the surrogate network and the sorting network, we can train the multi-head network. The algorithm is provided as pseudocode in the Appendix Algorithm 1.

## 4 EXPERIMENTS

Our experimental framework was applied to six datasets provided by OpenXAI (Agarwal et al., 2022), including both synthetic and empirical datasets, information summarized in the Appendix F Table 3. We utilized two pre-trained models from OpenXAI API for benchmarking: a logistic regressor (LR), an artificial neural network (ANN), and an interpretable decision tree (DT) for a more comprehensive comparison. The experimental design is structured to gain new insights into explanation disagreements and demonstrate how EXAGREE improves explanation agreements.

Table 1: Ground-truth and predictive faithfulness results ($k$=0.25) on the Adult Income dataset for all explanation methods with LR and ANN models. (↑) indicates that higher values are better, and (↓) indicates that lower values are better. Best values in each metric across explanation methods on each model are *italicized* and improved scores in SAEMs are in **bold**, applied to other datasets.

| | Method | FA(↑) | RA(↑) | SA(↑) | SRA(↑) | RC(↑) | PRA(↑) | PGI(↑) | PGU(↓) | #Best |
|---|---|---|---|---|---|---|---|---|---|---|
| | LIME | *1.00* | *1.00* | 0.00 | 0.00 | 0.99 | 0.99 | *0.15* | *0.04* | 4 |
| | SHAP | 0.50 | 0.25 | 0.00 | 0.00 | 0.48 | 0.69 | 0.08 | 0.13 | 0 |
| | Integrated Gradient | *1.00* | *1.00* | 0.00 | 0.00 | *1.00* | *1.00* | *0.15* | *0.04* | 6 |
| | Vanilla Gradient | *1.00* | *1.00* | 0.00 | 0.00 | *1.00* | *1.00* | *0.15* | *0.04* | 6 |
| LR | SmoothGrad | *1.00* | *1.00* | 0.00 | 0.00 | *1.00* | *1.00* | *0.15* | *0.04* | 6 |
| | Random | 0.75 | 0.00 | 0.50 | 0.00 | 0.18 | 0.55 | 0.13 | 0.06 | 0 |
| | Gradient x Input | 0.50 | 0.00 | 0.00 | 0.00 | 0.53 | 0.72 | 0.07 | 0.13 | 0 |
| | FIS_LR | 0.75 | 0.00 | 0.50 | 0.00 | 0.82 | 0.82 | 0.14 | 0.05 | 0 |
| | FIS_SAEM | ***1.00*** | **0.25** | **0.75** | **0.25** | **0.93** | **0.90** | ***0.15*** | ***0.04*** | 5 |
| | LIME | 0.50 | 0.25 | 0.00 | 0.00 | 0.74 | 0.74 | 0.23 | 0.06 | 0 |
| | SHAP | *0.75* | 0.00 | *0.50* | 0.00 | 0.75 | 0.81 | *0.24* | *0.06* | 4 |
| | Integrated Gradient | *0.75* | *0.75* | 0.00 | 0.00 | 0.65 | 0.72 | *0.24* | *0.06* | 4 |
| | Vanilla Gradient | 0.50 | 0.50 | 0.00 | 0.00 | 0.32 | 0.63 | 0.23 | 0.07 | 0 |
| ANN | SmoothGrad | 0.50 | 0.25 | 0.00 | 0.00 | 0.74 | 0.74 | 0.23 | *0.06* | 1 |
| | Random | *0.75* | 0.00 | *0.50* | 0.00 | 0.18 | 0.55 | *0.24* | 0.08 | 3 |
| | Gradient x Input | 0.25 | 0.00 | 0.00 | 0.00 | 0.06 | 0.53 | 0.06 | 0.24 | 0 |
| | FIS_ANN | *0.75* | 0.00 | *0.50* | 0.00 | *0.85* | 0.83 | *0.24* | 0.07 | 4 |
| | FIS_SAEM | *0.75* | **0.25** | *0.50* | **0.25** | *0.85* | **0.84** | *0.24* | ***0.06*** | 7 |
| | Decision Trees | 1.00 | 0.25 | 0.75 | 0.25 | 0.83 | 0.80 | 0.15 | 0.04 | - |

**Experimental Setup** To establish a consistent benchmark for stakeholder needs, we adopted the ground truth explanations derived from the pre-trained LR as our constant target ranking. This approach allows for systematic evaluation of explanation agreement across diverse scenarios, explanation methods, and models. OpenXAI built-in explanation methods (Agarwal et al., 2022) and Feature Importance Score (FIS) (Li et al., 2024) are used as feature attributions for pre-trained LR and ANN models. To quantify the agreement between various explanations and this established ground truth, we compared their feature importance rankings using the comprehensive set of evaluation metrics detailed in Sec. 2.3.

### 4.1 EXISTENCE OF EXPLANATION DISAGREEMENT PROBLEMS AND BEYOND

Explanation disagreement problems have been discussed and evidenced in previous studies Krishna et al. (2022); Li & Barnard (2024). Our aim here is not to reiterate these discussions, but to concisely formalize these problems within the context. To this end, we conducted a series of targeted

experiments across various models, explanation methods, and datasets, designed to illustrate these fundamental explanation disagreements.

Table 2: Ground-truth and predictive faithfulness results ($k$=0.25) on the Synthetic dataset for all explanation methods with LR and ANN models. (↑) indicates that higher values are better, and (↓) indicates that lower values are better.

|  | Method | FA(↑) | RA(↑) | SA(↑) | SRA(↑) | RC(↑) | PRA(↑) | PGI(↑) | PGU(↓) | #Best |
|---|---|---|---|---|---|---|---|---|---|---|
| LR | LIME | *1.00* | *1.00* | *1.00* | *1.00* | 0.99 | 0.98 | *0.13* | *0.07* | 6 |
|  | SHAP | *1.00* | 0.20 | 0.40 | 0.20 | 0.96 | 0.93 | *0.13* | *0.07* | 3 |
|  | Integrated Gradient | *1.00* | *1.00* | *1.00* | *1.00* | *1.00* | *1.00* | *0.13* | *0.07* | 8 |
|  | Vanilla Gradient | *1.00* | *1.00* | *1.00* | *1.00* | *1.00* | *1.00* | *0.13* | *0.07* | 8 |
|  | SmoothGrad | *1.00* | *1.00* | *1.00* | *1.00* | *1.00* | *1.00* | *0.13* | *0.07* | 8 |
|  | Random | 0.20 | 0.00 | 0.00 | 0.00 | -0.09 | 0.44 | 0.07 | 0.13 | 0 |
|  | Gradient x Input | 0.40 | 0.00 | 0.40 | 0.00 | 0.65 | 0.79 | 0.09 | 0.11 | 0 |
|  | FIS_LR | *1.00* | 0.60 | 0.40 | 0.20 | 0.97 | 0.94 | *0.13* | *0.07* | 3 |
|  | FIS_SAEM | *1.00* | **1.00** | 0.40 | **0.40** | 0.97 | 0.94 | *0.13* | *0.07* | 4 |
| ANN | LIME | 0.40 | 0.00 | 0.20 | 0.00 | 0.60 | 0.72 | 0.11 | 0.14 | 0 |
|  | SHAP | 0.60 | *0.20* | 0.00 | 0.00 | 0.65 | 0.74 | 0.11 | 0.13 | 1 |
|  | Integrated Gradient | *0.80* | *0.20* | 0.00 | 0.00 | 0.66 | 0.74 | 0.11 | 0.13 | 2 |
|  | Vanilla Gradient | 0.40 | *0.20* | *0.40* | *0.20* | 0.42 | 0.63 | *0.12* | 0.13 | 3 |
|  | SmoothGrad | 0.20 | 0.00 | 0.20 | 0.00 | 0.56 | 0.69 | 0.10 | 0.14 | 0 |
|  | Random | 0.20 | 0.00 | 0.00 | 0.00 | -0.09 | 0.44 | 0.06 | 0.16 | 0 |
|  | Gradient x Input | 0.40 | *0.20* | 0.20 | 0.00 | 0.47 | 0.66 | 0.11 | 0.13 | 1 |
|  | FIS_ANN | *0.80* | *0.20* | 0.20 | 0.00 | 0.79 | 0.74 | *0.12* | *0.12* | 4 |
|  | FIS_SAEM | *0.80* | *0.20* | 0.20 | 0.00 | **0.82** | **0.78** | *0.12* | *0.12* | 6 |
| | Decision Trees | 0.60 | 0.00 | 0.00 | 0.00 | 0.60 | 0.48 | 0.11 | 0.10 | - |

**Model Disagreement** We compared explanation agreements among the LR, ANN, and DT models for the same stakeholder needs, showing how explanation agreements can vary across different model architectures despite promising performance in Table 1, Table 2, and other Tables in the Appendix.

**Explanation Method Disagreement** We applied various explanation methods provided in OpenXAI (e.g., LIME, SHAP, Integrated Gradients), including common permutation-based and gradient-based methods, to the same model and stakeholder needs. The results highlight substantial variability in explanations across these different methods in Table 1, Table 2, and other Tables in the Appendix. It is also noted that well-established methods, such as SHAP might result in substantial disagreement.

**Ground Truth Disagreement** We compared intrinsic interpretations from DT, post-hoc explanations from both interpretable and black-box models (LR, ANN) against the assumed ground truth (LR coefficients), demonstrating disagreements among various metrics, shown in Table 1, Table 2, and other Tables in the Appendix.

### 4.1.1 NOTEWORTHY INSIGHTS INTO THE EXPLANATION DISAGREEMENT PROBLEMS

**Trends across Models, Explanation Methods, and Datasets** Our analysis reveals a trend: when stakeholder needs to align with the ground truth attribution of an interpretable model (LR in this case), explanation agreement (explaining LR itself) is generally high. This is evident in the LR results in Tables 1 and 2, where LIME and gradient-based methods show consistently high scores across FA, RA, RC, and PRA metrics. In contrast, when explaining models other than the ground truth model itself, regardless of their interpretability (e.g., ANN as a black-box model or DT as an interpretable model), explanation agreement often varies significantly across different datasets. For instance, DT shows perfect FA (1.0) in Adult Income but 0.6 in the Synthetic dataset (Tables 1 and 2).

From a broader perspective, explanation agreements derived from the ground truth model (LR) generally outperform those from black-box models (e.g., ANN) across most datasets. However, exceptions are observed in the COMPAS and GMSC datasets, where agreement levels between LR and black-box models are similar. This suggests that in these cases, the black-box models may coincidentally similarly make predictions as the ground truth model.

Some explanation methods exhibit better agreement in black-box models than the ground truth model. For instance, SHAP demonstrates higher agreement when explaining the ANN model, despite showing lower agreement with the LR model in Table 1. These observations highlight the complex

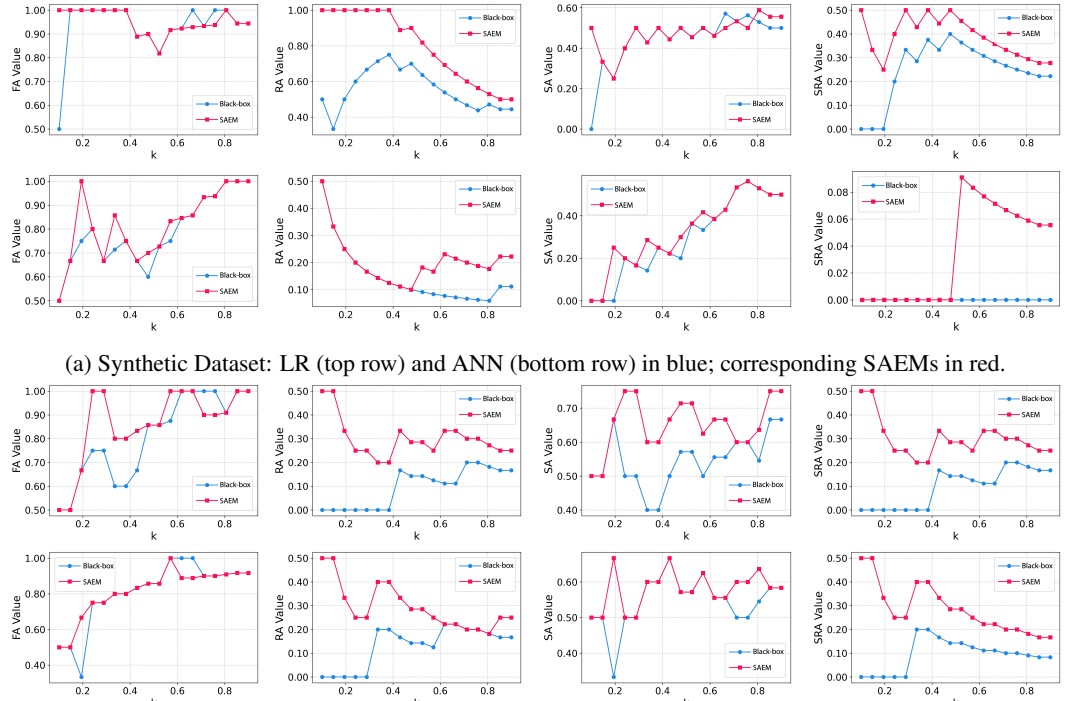

(a) Synthetic Dataset: LR (top row) and ANN (bottom row) in blue; corresponding SAEMs in red.

(b) Adult Income Dataset: LR (top row) and ANN (bottom row) in blue; corresponding SAEMs in red.

Figure 3: Comparison of faithfulness agreement metrics (FA, RA, SA, SRA) between assumed black box models (LR and ANN) and the identified SAEMs for varying $k$ values on the Synthetic dataset and Adult Income dataset.

nature of explanations and the challenges in finding a single model or explanation method that universally outperforms others, leading to the following discussion.

**The Significance of Stakeholder Disagreement** If we conceptualize different explanation methods (e.g., Random, LIME, and SHAP) as representing distinct stakeholder perspectives, stakeholder disagreements emerge from a single model (e.g., LR or ANN). Assuming LR coefficients as ground truth explanations, we might find that one stakeholder (represented by $s_{\text{LIME}}$) is satisfied, while another (represented by $s_{\text{SHAP}}$) is not. In such scenarios, we observe that greater disagreement between these "stakeholders" in one model (LR) can lead to potentially improved faithfulness for $s_{\text{SHAP}}$ in another model (ANN). In other words, $s_{\text{SHAP}}$ shows higher satisfaction with ANN than with LR. This observation reveals a crucial insight: when stakeholder requirements are diverse, relying on a single model often yields conflicting explanations, demonstrating stakeholder disagreement. It also underscores the potential existence of more faithful models capable of addressing previously unsatisfied stakeholder needs, showing the validity of Proposition in Sec. 3.2.1.

## 4.2 ENHANCING EXPLANATION AGREEMENT THROUGH EXAGREE

We now turn to demonstrate the efficacy of our EXAGREE framework in identifying models that enhance explanation agreement from the Rashomon set, with a focus on improving both faithfulness and fairness. EXAGREE was evaluated across six datasets, with the results partially presented in Tables 1 and 2, Figs. 3 and 4, as well as in additional tables and figures in the Appendix. An ablation study demonstrating the impact of $\epsilon$ on the Rashonmon set, and consequently on explanation agreements, is presented in the Appendix G Fig. 7.

**Faithfulness Analysis** The agreement metrics for the given models and the identified SAEMs are reported in Tables 1 and 2 for $k = 0.25$. This comparison allows us to evaluate SAEMs against other established explanation methods across the LR and ANN models. While the identified SAEMs do not always outperform other methods (e.g., LIME in the HELOC dataset in Table 9), they demonstrate superior agreement in most datasets compared with established explanation methods (e.g., SHAP).

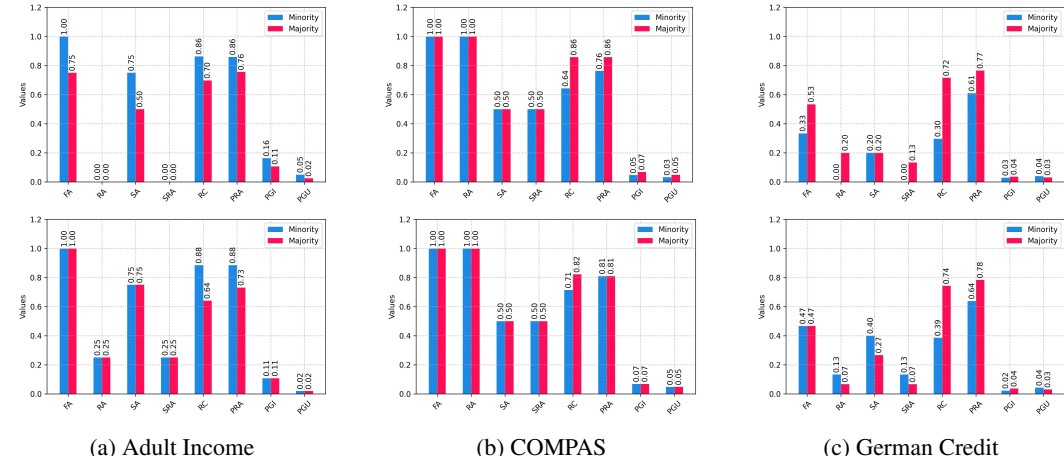

(a) Adult Income          (b) COMPAS          (c) German Credit

Figure 4: Comparison of fairness analysis between the LR model (top) and SAEM (bottom) for $k = 0.25$ on the Adult Income, COMPAS, and German Credit datasets. Faithfulness metrics are shown for majority (male, red) and minority (female, blue) subgroups. Larger gaps between subgroup values indicate higher, undesirable disparities.

More significantly, SAEMs consistently enhance explanation agreement relative to the provided models (assumed as black-box to stakeholders). This improvement is demonstrated through the visualizations of agreement metrics (FA, RA, SA, and SRA) across different $k$ values in Fig. 3.

**Fairness in Subgroups** Our fairness analysis, depicted in Fig. 4, compares the faithfulness between the pre-trained LR and the identified SAEM on three datasets that contain gender information (e.g., male and female). The provided model exhibits significant disparities in faithfulness metrics between majority and minority groups, indicating unfair explanations, particularly in the Adult Income dataset (see Fig. 4 (a)). The SAEMs identified by our framework reduce these inequities across all three datasets, showcasing an improvement in explanation fairness between subgroups.

**The Significance of Stakeholder-Centered Perspective** The significance of our framework goes beyond the performance metrics. By conceptualizing the LR ground truth explanations as a universal benchmark provided to all stakeholders and treating each method and model as distinct stakeholder perspectives (e.g., $s_{LR}$, $s_{ANN}$), we identify significant disagreements among stakeholders. The EXAGREE framework effectively enhances explanation agreement for both $s_{LR}$ and $s_{ANN}$, providing more fair and faithful explanations to the unique requirements of each stakeholder group.

**User-friendly Interface** The EXAGREE framework ultimately serves as a practical tool for diverse stakeholders. Therefore, we incorporate user-friendly functionality that bridges complex technical implementations and stakeholder needs by leveraging advancements in recent Large Language Models. EXAGREE the Gemini API (Google, 2023) to convert stakeholders' needs to attributions or rankings, allowing stakeholders to articulate preferences and domain knowledge without extensive machine learning expertise. We showcase a user case in the Appendix C Fig. 5.

## 5 CONCLUSION

In this work, we formalized the ranking-based explanation disagreement problem and advocated for a stakeholder-centered perspective that aims to meet specific needs. This approach lays a foundation for future studies in the field of explainable machine learning. To reconcile the complex challenge of explanation disagreement, we introduced EXAGREE, a novel framework that identifies SAEMs. By leveraging the Rashomon set concept, EXAGREE enhances both the faithfulness and fairness of model explanations, as demonstrated through rigorous experiments across various datasets. The dual improvement achieved by EXAGREE not only provides a practical solution adaptable to diverse stakeholder requirements but also effectively addresses subgroup fairness. This underscores the framework's potential to contribute to a more trustworthy and interpretable AI system in high-stakes decision-making contexts. Broader impact and limitations are discussed in the Appendix A.

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

## A    BROADER IMPACT AND LIMITATIONS

Our work highlights that no single model or explanation method can universally satisfy all stake-holders' needs. However, EXAGREE demonstrates that it's possible to identify more faithful and fair models for diverse stakeholders, meeting emerging AI fairness regulations The impact of this work could be substantial. By enhancing explanation agreement across different stakeholder groups, EXAGREE has the potential to increase trust in AI systems. This is particularly crucial in high-stakes domains such as healthcare, finance, and criminal justice, where the consequences of decisions are far-reaching and the need for trustworthy explanations is paramount. Furthermore, our findings highlight the need for deeper exploration of the interplay between Human-Computer Interaction (HCI) and Explainable AI (XAI). This intersection represents a rich area for future research, potentially leading to more user-centered and effective explainable AI systems.

We acknowledge several limitations of our current work: a) alternative approaches to Rashomon set sampling, sorting and ranking algorithms, and non-differentiable optimization could potentially enhance our framework. b) real-world scientific applications with diverse requirements would validate the framework's broader applicability. c) a more sophisticated interface could improve the framework's accessibility and usability for non-expert stakeholders. d) a more comprehensive and specific agreement evaluation metric framework can be further developed.

## B    RELATED WORK

Our work contributes to the expansive field of explainable artificial intelligence, with a specific focus on explainable machine learning Krishna et al. (2022). This section discusses closely related works and their connections to our framework, clarifying key terminologies and highlighting the ongoing challenges in the field. Given the interdisciplinary nature of our research, which spans areas including sorting and ranking, non-differential optimization, Rashomon sets, and human-centered interaction, we acknowledge that an exhaustive review of all related works is beyond the scope of this section. Instead, we concentrate on literature directly pertinent to our main objective: addressing the explanation disagreement problem. Other related areas are discussed briefly, as they serve as tools or methodologies to achieve this primary purpose.

### B.1    TERMINOLOGY AND CORE CONCEPTS

In the literature, the terms explanation/interpretation are often used interchangeably (Doshi-Velez & Kim, 2017; Lipton, 2018). Similarly, concepts such as explanation disagreement, inconsistency, and diversity all refer to scenarios where explanations differ, whether between models, methods, or human understanding (Krishna et al., 2022; Roscher et al., 2020). The problem of explanation disagreement remains a significant open challenge, hindering the impact of machine learning models (Krishna et al., 2022; Adebayo et al., 2018; Rudin, 2019; Ghassemi et al., 2021; Roscher et al., 2020; Ribeiro et al., 2016). This issue exists in various forms: when a single model generates different explanations, when similar-performing models produce distinct explanations, or when model explanations diverge from human expectations.

One of the most influential works from Rudin (2019) is using interpretable models instead of black box models to avoid the problem. The idea is naturally true. However, constructing interpretable models practically poses challenges for many groups (Adadi & Berrada, 2018). For instance, it is unrealistic for end users to construct an interpretable model before they ask for reasons behind their predicted results or find a model that meets all stakeholders' needs within a given context. More importantly, even though interpretable models, such as decision trees, generate rationale behind the prediction, such explanations are not always what stakeholders expected, formulated as ground truth disagreement. Both post-hoc for black-box models and ante-hoc for interpretable models - achieving consistent and reliable explanation agreement remains a challenge (Varshney & Alemzadeh, 2017; Jiménez-Luna et al., 2020; Huang et al., 2023; Zhong et al., 2022; Barnard & Fox, 2023; Barnard, 2022; Reichstein et al., 2019; Roscher et al., 2020). This leads to a common sense that ensuring human oversight of both predictions and their explanations is crucial for maintaining confidence in machine learning-assisted decision-making processes.

### B.2 OPTIMIZATION AND RANKING IN EXPLANATIONS

End-to-end optimization in machine learning often involves sorting and ranking operations, which present unique challenges due to their non-differentiable nature. Sorting is a piecewise linear function with numerous non-differentiable points, while ranking is a piecewise constant function with null or undefined derivatives. These properties make it difficult to incorporate sorting and ranking directly into gradient-based optimization frameworks.

Sorting networks, a concept dating back to the 19th century (Knuth, 1997), offer a potential solution. These highly parallel, data-oblivious sorting algorithms use conditional pairwise swap operators to map inputs to ordered outputs. Recent advancements have led to the development of differentiable sorting networks, also known as soft rank methods. These techniques approximate the discrete sorting operation with a continuous, differentiable function. One popular approach is the use of the neural sort operator (Grover et al., 2019), which employs a differentiable relaxation of the sorting operation. Another method involves using the optimal transport formulation to create a differentiable proxy for sorting (Cuturi et al., 2019). In the EXAGREE framework, we adopt the recent DiffSortNet algorithm proposed by Petersen et al. (2022). This approach offers several advantages, including the simplicity of using a logistic sigmoid function and the guarantee of monotonicity in the sorting operation. DiffSortNet provides a differentiable approximation of sorting that maintains the essential properties of traditional sorting while enabling gradient-based optimization. Sorting and ranking attributions in a Rashomon set also poses difficulties, we discussed the constraints in Sec. D

### B.3 RASHOMON SETS RELATED WORKS

Recent research has increasingly advocated for exploring sets of equally good models, rather than focusing on a single model Rudin (2019); Rudin et al. (2024); Hsu & Calmon (2022); Li et al. (2023). This approach, known as the Rashomon set concept, provides a more comprehensive understanding of model behavior and feature importance. Fisher et al. (2019) first introduced the concept of Model Class Reliance (MCR). Building on this work, Dong & Rudin (2020) explored the cloud of variable importance (VIC) for the set of all good models, providing concrete examples in linear and logistic regression. Further expanding on these ideas, Hsu & Calmon (2022) investigated instance-level explanations within a set of models, while Li et al. (2023) introduced the concept of Feature Interaction Score (FIS) in the Rashomon set. While these existing works have significantly advanced our understanding of model behavior through the lens of Rashomon sets, the practical benefits and applications of this approach have remained largely unexplored. Our work represents a significant step forward in this domain, as it presents the first practical application of the Rashomon set concept to address the challenge of explanation disagreement.

## C USER-FRIENDLY INTERFACE

The EXAGREE framework incorporates user-friendly functionality that bridges complex technical implementations and stakeholder needs. It accommodates various input forms, potentially enhancing its practical utility. Generally, stakeholders can directly provide expected feature attribution rankings, ensuring usability for diverse groups. Additionally, experts can specify detailed attributions or directional preferences for features. For instance, a stakeholder might express, "I believe this feature should have a negative impact on the outcome", or provide specific attribution values. Leveraging advancements in Large Language Models, EXAGREE utilizes natural language processing capabilities through the Gemini API (Google, 2023). This allows stakeholders to articulate preferences and domain knowledge without extensive machine learning expertise, making explainable AI more accessible in real-world applications. Here we provided a demo to illustrate the usage of these functionalities on an assumed stakeholder working with the provided black-box model ANN and the Synthetic dataset, as shown in Fig. 5.

## D CONSTRAINED SORTING AND RANKING IN A RASHOMON SET

Our primary objective of minimizing disagreement between model explanations and stakeholder expectations naturally leads to an end-to-end ranking optimization problem. The Rashomon set concept allows us to transform single attribution values into ranges, as illustrated in Fig. 6, enabling

Given the provided model ann and dataset gaussian, could you please share your preferences for feature importance from the feature list ['gauss_0', 'gauss_1', 'gauss_2', 'gauss_3', 'gauss_4', 'gauss_5', 'gauss_6', 'gauss_7', 'gauss_8', 'gauss_9', 'gauss_10', 'gauss_11', 'gauss_12', 'gauss_13', 'gauss_14', 'gauss_15', 'gauss_16', 'gauss_17', 'gauss_18', 'gauss_19']?

(a) The welcome message from our demo GUI

I hope gauss_0 is less important than gauss_1 and gauss_1 is then less important than gauss_2, I also hope gauss_4 is more important then gauss_3 and gauss_3 is less important than gauss_0.

Send

(b) The prompt regarding to guass_0, guass_1, guass_2, guass_3 and guass_4.

The degree of agreement between your desired feature ranking and the given model's behavior, focusing on the top 50% most important features, is as follows:
Agreement Metrics:
- Feature Agreement: 0.60
- Rank Agreement: 0.00
- Sign Agreement: 0.40
- Sign Rank Agreement: 0.00
- Pairwise Rank Agreement: 0.59
- Rank Correlation: 0.61

Are you happy with the explanation? (Yes/No)

(c) Initial response of calculated agreement: FA:0.6; RA:0.0; SA:0.4; SRA:0

Based on your preferences, I've analyzed the feature importance. Here's a model that aligns well with your expectations:

Agreement Metrics:
- Feature Agreement: 0.70
- Rank Agreement: 0.20
- Sign Agreement: 0.40
- Sign Rank Agreement: 0.20
- Pairwise Rank Agreement: 0.59
- Rank Correlation: 0.60

(d) Second response from identified SAEM: FA:0.7; RA:0.2; SA:0.4; SRA:0.2

Figure 5: Demonstration of EXAGREE's user-friendly interface for improving explanation agreement aligned with stakeholder preferences. (a) The system prompts the user to provide feature importance preferences for a given ann (black-box model) and synthetic dataset. (b) The user expresses preferences using natural language, specifying the relative importance of five features. (c) EXAGREE evaluates the agreement between the user's desired feature ranking and the current model's behavior, presenting various agreement metrics. (d) The user is not happy with the outcome. After optimization, EXAGREE presents an improved model that better aligns with the user's expectations, showing enhanced agreement metrics. This interactive process allows stakeholders to shape model explanations without requiring deep technical expertise.

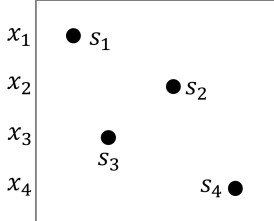 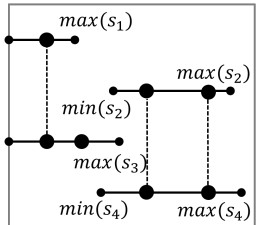 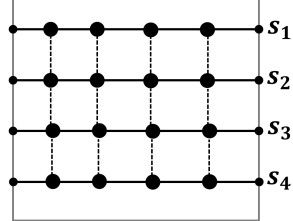

Figure 6: An illustration of feature attributions in a single model and a model space. Left: feature attributions from a single model using an arbitrary explanation method; Middle: feature attributions in a Rashomon set; Right: a conventional ranking problem setting.

us to find a potential model with stakeholder-expected rankings from $\mathcal{M}_\epsilon$. However, this optimization is constrained within the Rashomon set—a collection of similarly performing models—as not all attribution swaps are feasible. This constraint distinguishes our problem from conventional ranking problems and necessitates a novel approach.

**Lemma.** *In a Rashomon set $\mathcal{M}_\epsilon$, not all pairwise attribution swaps are possible.*

*Proof.* Assume, for the sake of contradiction, that any pairwise swap of attributions is possible within the Rashomon set. However, this contradicts the fact that the feature attribution range or model class reliance is not unlimited, as demonstrated in prior Rashomon-related theoretical and empirical studies (Fisher et al., 2019; Li et al., 2023; Hsu & Calmon, 2022; Xin et al., 2022; Li et al., 2024). Consider a scenario where the feature attribution matrix from the Rashomon set is:

$$[min(\mathbf{a}_i), max(\mathbf{a}_i)]_{i=1}^p,$$

where we specify $max(a_1) < min(a_2)$, as illustrated in the middle panel of Fig. 6. In this case, the swap $a_{(1)}$ between $a_{(2)}$ is impossible within this Rashomon set. $\square$

**Proposition.** *Based on the previous lemma, we can conclude that there does not always exist a model within the Rashomon set $\mathcal{M}_\epsilon$ that satisfies a stakeholder's expectation.*

*Proof.* As shown in the lemma, consider an expected ranking is $\mathbf{r}^* = (a_{(1)} \succ a_{(2)} \succ \cdots \succ a_{(p)})$, and the feature attribution matrix from the Rashomon set exhibits the property $max(a_1) < min(a_2)$, there does not exist a model $M \in \mathcal{M}_\epsilon$ that can satisfy the condition $a_{(1)} \succ a_{(2)}$, formulated as:

$$\nexists M \in \mathcal{M}_\epsilon : a_{(1)} \succ a_{(2)}.$$

This means that the stakeholder's expected ranking $\mathbf{r}^*$ is inaccessible within this Rashomon set $\mathcal{M}_\epsilon$, as the necessary pairwise attribution swaps to achieve the desired ranking order is not feasible. $\square$

Given this limitation, our approach shifts towards finding a more faithful model within the Rashomon set that better aligns with stakeholder expectations, even if it doesn't perfectly match them. This insight motivates the development of our multi-component framework.

# E   ADDITIONAL PROOF

**Lemma.** *For a given target ranking $\mathbf{r}^*$, there may exist multiple distinct rankings that have the same Spearman's rank correlation coefficient with the target ranking.*

*Proof.* The Spearman's rank correlation coefficient $\rho$ between any ranking $\mathbf{r}$ and the target ranking $\mathbf{r}^*$ can be expressed as:

$$\rho^*(\mathbf{r}, \mathbf{r}^*) = 1 - \frac{6\sum_{i=1}^n d_i^2}{n(n^2-1)},$$

where $d_i = r_i^* - r_i$ is the difference between the ranks of the $i$-th element in the target ranking $\mathbf{r}^*$ and the compared ranking $\mathbf{r}$, and $n$ is the total number of elements in the ranking. This formula shows that $\rho$ depends solely on the sum of squared rank differences $D = \sum_{i=1}^n d_i^2$.

Let us define two distinct rankings: $\mathbf{r} = [r_1, r_2, \ldots, r_n]$ and $\mathbf{r}' = [r_1', r_2', \ldots, r_n']$. The rank difference for the $i$-th element in relation to the target ranking is given by: $d_i = r_i^* - r_i$, and $d_i' = r_i^* - r_i'$. Thus, The sum of squared rank differences for each ranking in $\mathbf{r}$ is expressed as:

$$D_r = \sum_{i=1}^{n} d_i^2 = \sum_{i=1}^{n} (r_i^* - r_i)^2,$$

and for $\mathbf{r}'$:

$$D_{r'} = \sum_{i=1}^{n} d_i'^2 = \sum_{i=1}^{n} (r_i^* - r_i')^2.$$

To establish that $D_r = D_{r'}$, we expand the expressions:

$$D_r = \sum_{i=1}^{n} (r_i^* - r_i)^2 = \sum_{i=1}^{n} (r_i^{*2} - 2r_i^* r_i + r_i^2) \quad D_{r'} = \sum_{i=1}^{n} (r_i^* - r_i')^2 = \sum_{i=1}^{n} (r_i^{*2} - 2r_i^* r_i' + r_i'^2).$$

Setting $D_r = D_{r'}$ leads to:

$$\sum_{i=1}^{n} (r_i^{*2} - 2r_i^* r_i + r_i^2) = \sum_{i=1}^{n} (r_i^{*2} - 2r_i^* r_i' + r_i'^2).$$

Cancelling $\sum_{i=1}^{n} r_i^{*2}$ from both sides gives:

$$\sum_{i=1}^{n} (-2r_i^* r_i + r_i^2) = \sum_{i=1}^{n} (-2r_i^* r_i' + r_i'^2).$$

Rearranging, we find that for $D_r = D_{r'}$, the following condition must hold:

$$\sum_{i=1}^{n} (r_i^2 - r_i'^2) = 2 \sum_{i=1}^{n} r_i^* (r_i' - r_i).$$

This implies that either: 1. The differences between $r_i'$ and $r_i$ must balance out when weighted by the corresponding $r_i^*$. 2. The squared values of the ranks in $r_i$ and $r_i'$ must differ in a way that maintains the overall relationship with $r_i^*$. These conditions are not mutually exclusive and can be satisfied simultaneously. They allow for the existence of distinct rankings $\mathbf{r}$ and $\mathbf{r}'$ that maintain the same Spearman's rank correlation with $\mathbf{r}^*$.

**In the case** $\rho = 1$ **or** $\rho = -1$, there are no distinct rankings, such as $\mathbf{r}$ and $\mathbf{r}'$ that can yield the same sum of squared rank differences $D_r = D_{r'}$, as any deviation would introduce non-zero differences, violating the condition $D_r = D_{r'}$.

**In the case** $\rho \in (-1, 1)$, the constraints are relaxed, allowing for the possibility of multiple distinct rankings $r$ and $r'$ yielding the same $D$. The possibility of such cases increases with a greater number of elements $n$, as the number of distinct permutations that can maintain the same squared rank differences increases. Thus, the condition for two distinct rankings $\mathbf{r}$ and $\mathbf{r}'$ to have the same sum of squared rank differences $D$ with respect to a target ranking $\mathbf{r}^*$ is established as described above.

Therefore, multiple distinct rankings can achieve the same Spearman's rank correlation coefficient with the target ranking $\mathbf{r}^*$, which completes the proof. □

**Remark 5.** *One example is: let* $\mathbf{r}^* = [1, 2, 3, 4, 5]$ *be our target ranking. Consider the following two distinct rankings:* $\mathbf{r} = [1, 3, 2, 5, 4]$ *and* $\mathbf{r}' = [2, 1, 3, 5, 4]$, *it is easy to calculate their correlations* $\rho(\mathbf{r}^*, \mathbf{r}) = \rho(\mathbf{r}^*, \mathbf{r}') = 0.8$, *and show the equivalence.*

**Proposition.** *The increase in disagreement among stakeholders leads to greater opportunity to find a more faithful model, as expressed by:*

$$\mathbb{P}\left( \exists M^* \in \mathcal{M}, \rho(\mathbf{r}^{M,\varphi}, \mathbf{r}^j) < \rho(\mathbf{r}^{M^*,\varphi}, \mathbf{r}^j) \right) \propto (1 - \rho(\mathbf{r}^i, \mathbf{r}^j)), \tag{8}$$

*where* $1 - \rho(\mathbf{r}^i, \mathbf{r}^j)$ *is the disagreement between two stakeholders* $i$ *and* $j$ *and* $i \neq j$.

*Proof.* Assume that stakeholder $i$ is fully satisfied by the current model $M$, which means $\rho(\mathbf{r}^{M,\varphi}, \mathbf{r}^i) = 1$. This condition implies that the ranking produced by the model $M$ perfectly matches stakeholder $i$'s expected ranking, i.e., $\mathbf{r}^{M,\varphi} = \mathbf{r}^i$.

Given the disagreement between stakeholders $i$ and $j$, we know that $\mathbf{r}^i \neq \mathbf{r}^j$, and therefore $\rho(\mathbf{r}^{M,\varphi}, \mathbf{r}^j) < 1$. As the level of disagreement between the stakeholders increases, represented by a larger value of $1 - \rho(\mathbf{r}^i, \mathbf{r}^j)$, the correlation $\rho(\mathbf{r}^{M,\varphi}, \mathbf{r}^j)$ correspondingly decreases.

Since the expected ranking $\mathbf{r}^j$ of stakeholder $j$ is fixed, an increase in disagreement effectively broadens the search space for potential models $M^*$. This broader search space implies that there are more opportunities to find a model $M^*$ such that $\rho(\mathbf{r}^{M^*,\varphi}, \mathbf{r}^j) > \rho(\mathbf{r}^{M,\varphi}, \mathbf{r}^j)$, thereby achieving an enhanced agreement for stakeholder $j$. This completes the proof. $\square$

## F  SUMMARY OF INFORMATION

We summarized notations used in the work, detained datasets information, pseudocode of the algorithm, and evaluation metrics in this section.

Table 3: Summary of Evaluation Metrics

| Metric | Description |
| --- | --- |
| Feature Agreement (FA) | Measures agreement in feature importance |
| Rank Agreement (RA) | Assesses agreement in feature ranking |
| Sign Agreement (SA) | Evaluates agreement in feature attribution signs |
| Signed Rank Agreement (SRA) | Combines sign and rank agreement |
| Pairwise Rank Agreement (PRA) | Measures pairwise rank consistency |
| Rank Correlation (RC) | Quantifies correlation between feature rankings |
| Prediction Gap on Important feature perturbation (PGI) | Measures Impact of perturbing important features on model predictions |
| Prediction Gap on Unimportant feature perturbation (PGU) | Measures impact of perturbing unimportant features on model predictions |
| Fairness | Compares all above metric values across majority and minority subgroups |

### F.1  HYPERPARAMETER AND MODEL STRUCTURE SUMMARY

We used pre-trained LR and ANN models from OpenXAI (Agarwal et al., 2022). The GRS sampling used an epsilon rate of 0.05 with log loss (Li et al., 2024). Our DMAN used two hidden layers of 100 units each, with input and output sizes matching the feature-length. The DiffSortNet used a bitonic sorting network with Cauchy interpolation. The MHMN employed 50 heads. Both DMAN and MHMN used Adam optimizer with learning rate scheduling. We also included a simple DT classifier information for comparison. An overall summary of hyperparameters is provided in Table 4. It is important to note that while this configuration yielded robust results across most of our tested datasets, it may not universally produce optimal outcomes for all scenarios. We provide these parameters as a strong starting point for further fine-tuning and adaptation to specific use cases.

## G  ABLATION STUDY

This study investigates how the size of the Rashomon set, controlled by the parameter $\epsilon$, affects EXAGREE's ability to identify models that align with stakeholder expectations. We conducted experiments on the Synthetic dataset using both pre-trained LR and ANN models provided by OpenXAI, varying $\epsilon$ values (0.05, 0.1, and 0.2). Intuitively, a larger Rashomon set should provide more opportunities for finding stakeholder-aligned models due to a larger search space.

**Results and Discussion**: As illustrated in Fig. 7, increasing $\epsilon$ generally enhances EXAGREE's capacity to identify models with improved explanation agreement. This is evidenced by consistent improvements in FA and RA metrics across all $k$ values as $\epsilon$ increases for both LR and ANN models.

---

**Algorithm 1** EXAGREE Framework Pseudocode (from stage 2)

---

**Require:** A task $\mathcal{D}_{task}$ and different requirements $\{i \in \mathcal{S} \mid \mathbf{r}^i\}$ from stakeholders $\mathcal{S}$.
1: Initialize pre-trained $f_{ref}$, $f_{dman}$, and $f_{diffsort}$.
2: **for** $i = 0, 1, 2, \cdots$ **do**          ▷ Find solutions for each stakeholder group
3:      $\mathbf{r}^{target} \leftarrow f_{diffsort}(|\mathbf{a}^i|)$, $\text{sign}(\mathbf{a}^i) \leftarrow \mathbf{a}^i$
4:      $\mathbf{r}^{ref} \leftarrow f_{diffsort}(|\mathbf{a}|)$          ▷ Compute target and reference rankings
5:      $\mathbf{p}_{top\_k} \leftarrow \text{Sort}(\mathbf{r}^{ref}, k)$
6:      $\mathbf{p}_{diff\_k} \leftarrow \text{Diff}(\mathbf{r}^{ref}, \mathbf{r}^{target}, k)$      ▷ Identify important features that make differences
7:      $\mathbf{M} \leftarrow \text{initialize\_mask}(h, \mathbf{p}_{top\_k}, \mathbf{p}_{diff\_k})$      ▷ Initialize $h$ masks with attentions
8:      ▷ Note: each head corresponds to a mask, and all masks are validated in the initialization
9:      **for** epoch $= 0, 1, \cdots, num\_epochs$ **do**
10:          $\mathbf{M}.\text{update}()$      ▷ Update all masks with states and keep a record of valid masks
11:          $\mathbf{i}_{active} \leftarrow \mathbf{M}.states$          ▷ Ensure masks within the $\mathcal{R}_\epsilon$
12:          **for** $\mathbf{m} \in \mathbf{M}_{active}$ **do**          ▷ Update the active masks only
13:              $\mathbf{a} \leftarrow f_{dman}(\mathbf{m})$          ▷ Approximate attributions of features
14:              $\mathbf{r}^{\mathbf{m},\varphi} \leftarrow f_{diffsort}(|\mathbf{a}|)$, $\text{sign}(\mathbf{a}) \leftarrow \mathbf{a}$      ▷ Calculate the ranking from DiffSortNet
15:              $l_{rank} \leftarrow \text{spearman\_cor}(\mathbf{r}^{\mathbf{m},\varphi}, \mathbf{r}^{target})$      ▷ Calculate the Spearman's correlation
16:              $l_{sign} \leftarrow \text{MSE}(\text{sign}(\mathbf{a}), \text{sign}(\mathbf{a}^i))$          ▷ Calculate loss on sign
17:              $\mathbf{l}_{rank}.\text{append}(l_{rank})$
18:              $\mathbf{l}_{sign}.\text{append}(l_{sign})$
19:          $\mathcal{L}_{rank} \leftarrow \text{Avg}(\mathbf{l}_{rank})$, $\mathcal{L}_{sign} \leftarrow \text{Avg}(\mathbf{l}_{sign})$
20:          $\mathcal{L}_{sparsity}, \mathcal{L}_{diversity} \leftarrow \text{Avg}(\text{norm}(\mathbf{M}_{active}), \text{dim=0}), \text{Avg}(\text{norm}(\mathbf{M}_{active}), \text{dim=1}))$
21:          $\mathcal{L}_{total} \leftarrow \mathcal{L}_{rank} + \mathcal{L}_{sign} + \lambda_s \times \mathcal{L}_{sparsity} + \lambda_d \times \mathcal{L}_{diversity}$
22:          $\mathcal{L}_{total}.\text{backward}()$
23:          $\mathbf{M}_{active}.\text{update}()$          ▷ Update active masks only
24:      $\mathbf{M}_{saved}.\text{evaluate}()$ ▷ Calculate true attributions and evaluate according to OpenXAI metrics
25:      $\mathbf{m}^i = \mathbf{M}_{saved}.\text{solution}()$
26: ▷ Found model $M = \mathbf{m}^i \circ f_{ref}$ that improve the overall faithfulness for the stakeholder group $i$
27:

---

Table 4: An Overall Summary of Hyperparameters in Experiments

| Component | Parameter | Value |
|---|---|---|
| GRS | Epsilon rate ($\epsilon$)
Loss function | 0.05, (0.1, 0.2 in Synthetic Dataset)
Log loss |
| DMAN | Model structure
Optimizer | [n_features, 100, 100, n_features]
Adam ($1 \times 10^{-4}$) |
| DiffSortNet | Type = Bitonic, Steepness = 10, Interpolation = Cauchy, ART lambda ($\lambda$) = 0.25 | |
| MHMN | Number of heads ($h$)
Optimizer
LR scheduler step size
LR scheduler gamma
Diversity weight & Sparsity weight | 50
Adam (0.01)
50
0.5
0.1 |
| Decision Tree | random state = 0, Max depth = 5, Min samples leaf = 5, Min samples split = 5 | |

Table 5: Summary of Datasets Information from OpenXAI

| Dataset | Size | # Features | Feature Types | Feature Information | Balanced |
|---|---|---|---|---|---|
| Synthetic Data | 5,000 | 20 | continuous | synthetic | True |
| German Credit | 1,000 | 20 | discrete, continuous | demographic, personal, financial | False |
| HELOC | 9,871 | 23 | continuous | demographic, financial | True |
| Adult Income | 45,222 | 13 | discrete, continuous | demographic, personal, education/employment, financial | False |
| COMPAS | 6,172 | 7 | discrete, continuous | demographic, personal, criminal | False |
| Give Me Some Credit | 102,209 | 10 | discrete, continuous | demographic, personal, financial | False |

However, the results for SA and SRA are not substantially improved: For the LR model, SA and SRA show mixed results, with improvements at certain $k$ values but not others. For the ANN model, all metrics, including SA and SRA, demonstrate clear improvements as $\epsilon$ increases from 0.05 to 0.2, except SRA when $\epsilon = 0.1$.

These findings generally support the hypothesis that larger Rashomon sets facilitate better stakeholder alignment. However, the lack of improvement in some cases (e.g., SRA in Fig. 7 (a)) aligns with the work of Li et al. (2024), which suggests that a larger Rashomon set does not necessarily guarantee a greater range of feature attributions. This observation potentially explains the uneven improvements in agreement metrics. In practice, decision-makers must carefully consider the trade-off between performance tolerance (determined by $\epsilon$) and the increased opportunities to find SAEMs.

# H  ADDITIONAL RESULTS

In this section, we reported additional predictive faithfulness results ($k$=0.25) from datasets: COMPAS, German Credit, HELOC, and Give Me Some Credit. Gradient-based explanations consistently demonstrate excellent agreement on LR models across all datasets, aligning with expectations given that the ground truth explanations are derived from LR coefficients. Interestingly, when applied to ANNs, some of these methods also show promising agreement, suggesting that in certain cases, these black-box models may coincidentally make predictions in a manner similar to the ground truth LR model. However, when considering DT as the ground truth, explanations derived from LR exhibit significantly varying levels of agreement. This variability underscores the critical importance of aligning explanation methods with stakeholder needs and model structures, instead of trusting explanations blindly.

Table 6: Comprehensive Notation Summary for EXAGREE Framework

| Notation | Description |
|---|---|
| $x, y, z$ | Scalars |
| $\mathbf{v}, \mathbf{w}$ | Vectors |
| $\mathbf{A}, \mathbf{B}$ | Matrices |
| $\mathcal{R}, \mathcal{Q}, \mathcal{D}$ | Sets |
| $v_i$ | $i$-th element of vector $\mathbf{v}$ |
| $a_{ij}$ | Element in $i$-th row and $j$-th column of matrix $\mathbf{A}$ |
| $\mathbf{a}_{i\cdot}$ | $i$-th row of matrix $\mathbf{A}$ |
| $\mathbf{a}_{\cdot j}$ | $j$-th column of matrix $\mathbf{A}$ |
| $(\mathbf{X}, \mathbf{y})$ | Dataset in $\mathbb{R}^{n \times (p+1)}$ |
| $\mathbf{X}$ | Covariate input matrix in $\mathbb{R}^{n \times p}$ |
| $\mathbf{y}$ | Output vector in $\mathbb{R}^n$ |
| $n$ | Number of instances |
| $p$ | Number of features |
| $h$ | Number of heads |
| $\mathcal{M}$ | Set of all considered models |
| $\Phi$ | Set of explanation methods |
| $\mathcal{S}$ | Set of stakeholders $\{s_1, s_2, s_3, \cdots\}$ |
| $\mathcal{F}$ | Set of features $\{\mathbf{x}_{\cdot 1}, \mathbf{x}_{\cdot 2}, \ldots, \mathbf{x}_{\cdot p}\}$ |
| $\mathcal{M}_\mathcal{I}$ | Subset of interpretable models |
| $f$ | Predictive model: $\mathbb{R}^{n \times p} \to \mathbb{R}^n$ |
| $\mathcal{L}$ | Loss function: $\mathbb{R}^n \times \mathbb{R}^n \to \mathbb{R}$ |
| $\mathbf{X}_{\backslash i}$ | Input matrix with $i$-th feature replaced |
| $a_i$ | Attribution measure for feature $i$ |
| $\mathbf{a}$ | Vector of attributions for all features $\{a_0, a_1, \cdots, a_p\}$ |
| $\mathbf{r}$ | Ranking of features based on attributions $\{r_0, r_1, \cdots, r_p\}$ |
| $\mathbf{a}^{M,\varphi}$ | Attribution vector for model $M$ and explanation method $\varphi$ |
| $\mathbf{r}^{M,\varphi}$ | Feature ranking for model $M$ and explanation method $\varphi$ |
| $\mathbf{a}^{M_\mathcal{I}}_{\text{true}}$ | Ground truth attribution vector for interpretable model $M_\mathcal{I}$ |
| $\mathbf{r}^{M_\mathcal{I}}_{\text{true}}$ | Ground truth ranking for interpretable model $M_\mathcal{I}$ |

Table 7: Ground-truth and predictive faithfulness results ($k$=0.25) on the COMPAS dataset for all explanation methods with LR and ANN models. ($\uparrow$) indicates that higher values are better, and ($\downarrow$) indicates that lower values are better.

| | Method | FA($\uparrow$) | RA($\uparrow$) | SA($\uparrow$) | SRA($\uparrow$) | RC($\uparrow$) | PRA($\uparrow$) | PGI($\uparrow$) | PGU($\downarrow$) | #Best |
|---|---|---|---|---|---|---|---|---|---|---|
| | LIME | *1.00* | *1.00* | *1.00* | *1.00* | *1.00* | *1.00* | *0.06* | *0.05* | 8 |
| | SHAP | 0.50 | 0.00 | 0.50 | 0.00 | 0.11 | 0.57 | 0.05 | 0.06 | 0 |
| | Integrated Gradient | *1.00* | *1.00* | *1.00* | *1.00* | *1.00* | *1.00* | *0.06* | *0.05* | 8 |
| | Vanilla Gradient | *1.00* | *1.00* | *1.00* | *1.00* | *1.00* | *1.00* | *0.06* | *0.05* | 8 |
| LR | SmoothGrad | *1.00* | *1.00* | *1.00* | *1.00* | *1.00* | *1.00* | *0.06* | *0.05* | 8 |
| | Random | 0.50 | 0.50 | 0.50 | 0.50 | 0.79 | 0.76 | 0.05 | 0.06 | 0 |
| | Gradient x Input | 0.50 | 0.00 | 0.50 | 0.00 | -0.04 | 0.52 | 0.05 | 0.06 | 0 |
| | FIS_LR | *1.00* | *1.00* | *0.50* | *0.50* | 0.86 | 0.86 | *0.06* | *0.05* | 5 |
| | FIS_SAEM | *1.00* | *1.00* | *0.50* | *0.50* | 0.89 | 0.90 | ***0.06*** | ***0.05*** | 5 |
| | LIME | *1.00* | 0.00 | *1.00* | 0.00 | 0.86 | 0.86 | *0.10* | *0.03* | 4 |
| | SHAP | 0.50 | 0.00 | 0.50 | 0.00 | -0.04 | 0.52 | 0.07 | 0.07 | 0 |
| | Integrated Gradient | *1.00* | 0.00 | 0.00 | 0.00 | 0.64 | 0.71 | *0.10* | *0.03* | 3 |
| | Vanilla Gradient | *1.00* | 0.00 | *1.00* | 0.00 | 0.82 | 0.81 | *0.10* | *0.03* | 4 |
| ANN | SmoothGrad | *1.00* | 0.00 | *1.00* | 0.00 | 0.86 | 0.86 | *0.10* | *0.03* | 4 |
| | Random | 0.50 | 0.50 | 0.50 | *0.50* | 0.79 | 0.76 | 0.08 | 0.07 | 1 |
| | Gradient x Input | 0.00 | 0.00 | 0.00 | 0.00 | -0.39 | 0.33 | 0.01 | 0.10 | 0 |
| | FIS_LR | *1.00* | *1.00* | *0.50* | *0.50* | 0.86 | 0.86 | *0.10* | *0.03* | 5 |
| | FIS_SAEM | *1.00* | *1.00* | *0.50* | *0.50* | ***0.95*** | ***0.96*** | *0.10* | *0.03* | 7 |
| | Decision Trees | 1.00 | 1.00 | 0.50 | 0.50 | 0.95 | 0.95 | 0.06 | 0.05 | - |

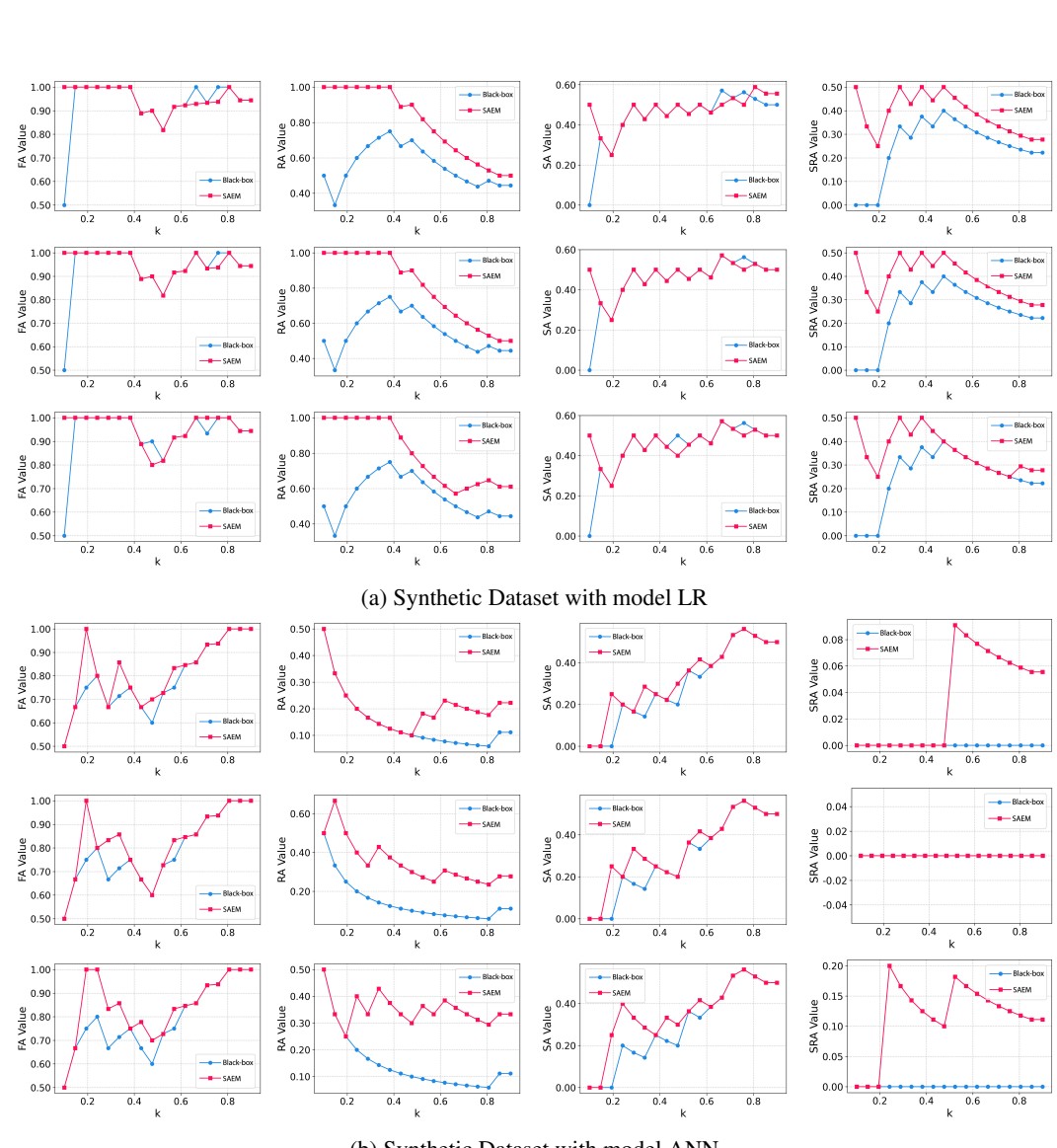

Figure 7: Comparison of faithfulness agreement metrics (FA, RA, SA, SRA) between black box models and their corresponding explanation-expected models for varying $\epsilon$ values on the Synthetic dataset. From top to bottom, $\epsilon$ is 0.05, 0.1, and 0.2, respectively.

Table 8: Ground-truth and predictive faithfulness results ($k$=0.25) on the German Credit dataset for all explanation methods with LR and ANN models. (↑) indicates that higher values are better, and (↓) indicates that lower values are better.

| | Method | FA(↑) | RA(↑) | SA(↑) | SRA(↑) | RC(↑) | PRA(↑) | PGI(↑) | PGU(↓) | #Best |
|---|---|---|---|---|---|---|---|---|---|---|
| | LIME | *1.00* | 0.87 | *1.00* | 0.87 | *1.00* | *1.00* | *0.04* | *0.03* | 6 |
| | SHAP | 0.53 | 0.00 | 0.53 | 0.00 | 0.49 | 0.68 | *0.04* | *0.03* | 2 |
| | Integrated Gradient | *1.00* | *1.00* | *1.00* | *1.00* | *1.00* | *1.00* | *0.04* | *0.03* | 8 |
| | Vanilla Gradient | *1.00* | *1.00* | *1.00* | *1.00* | *1.00* | *1.00* | *0.04* | *0.03* | 8 |
| LR | SmoothGrad | *1.00* | *1.00* | *1.00* | *1.00* | *1.00* | *1.00* | *0.04* | *0.03* | 8 |
| | Random | 0.27 | 0.00 | 0.13 | 0.00 | -0.15 | 0.45 | 0.02 | 0.05 | 0 |
| | Gradient x Input | 0.53 | 0.00 | 0.53 | 0.00 | 0.47 | 0.67 | *0.04* | *0.03* | 2 |
| | FIS_LR | 0.53 | 0.07 | 0.33 | 0.07 | 0.70 | 0.76 | *0.04* | *0.03* | 2 |
| | FIS_SAEM | 0.60 | 0.07 | 0.40 | 0.07 | 0.68 | 0.76 | **0.04** | **0.03** | 2 |
| | LIME | 0.40 | 0.00 | 0.40 | 0.00 | 0.36 | 0.62 | 0.05 | *0.10* | 1 |
| | SHAP | 0.33 | 0.00 | 0.33 | 0.00 | -0.02 | 0.49 | *0.07* | *0.10* | 2 |
| | Integrated Gradient | 0.40 | 0.00 | 0.40 | 0.00 | 0.27 | 0.59 | 0.06 | *0.10* | 1 |
| | Vanilla Gradient | 0.33 | 0.00 | 0.33 | 0.00 | *0.38* | *0.63* | 0.06 | *0.10* | 3 |
| ANN | SmoothGrad | 0.33 | 0.00 | 0.33 | 0.00 | 0.35 | 0.62 | 0.06 | *0.10* | 1 |
| | Random | 0.27 | 0.00 | 0.13 | 0.00 | -0.15 | 0.45 | 0.05 | 0.11 | 0 |
| | Gradient x Input | *0.47* | 0.00 | *0.47* | 0.00 | 0.08 | 0.52 | *0.07* | *0.10* | 4 |
| | FIS_ANN | 0.40 | 0.07 | 0.26 | 0.07 | 0.06 | 0.52 | 0.06 | *0.10* | 1 |
| | FIS_SAEM | **0.47** | **0.13** | **0.47** | **0.13** | 0.05 | 0.52 | 0.06 | *0.10* | 5 |
| | Decision Trees | 0.27 | 0.00 | 0.13 | 0.00 | 0.21 | 0.48 | 0.04 | 0.03 | - |

Table 9: Ground-truth and predictive faithfulness results ($k$=0.25) on the HELOC dataset for all explanation methods with LR and ANN models. (↑) indicates that higher values are better, and (↓) indicates that lower values are better.

| | Method | FA(↑) | RA(↑) | SA(↑) | SRA(↑) | RC(↑) | PRA(↑) | PGI(↑) | PGU(↓) | #Best |
|---|---|---|---|---|---|---|---|---|---|---|
| | LIME | *1.00* | *1.00* | *1.00* | *1.00* | 0.99 | 0.97 | *0.09* | *0.05* | 6 |
| | SHAP | 0.33 | 0.17 | 0.17 | 0.00 | 0.57 | 0.68 | 0.06 | 0.09 | 0 |
| | Integrated Gradient | *1.00* | *1.00* | *1.00* | *1.00* | *1.00* | *1.00* | *0.09* | *0.05* | 8 |
| | Vanilla Gradient | *1.00* | *1.00* | *1.00* | *1.00* | *1.00* | *1.00* | *0.09* | *0.05* | 8 |
| LR | SmoothGrad | *1.00* | *1.00* | *1.00* | *1.00* | *1.00* | *1.00* | *0.09* | *0.05* | 8 |
| | Random | 0.17 | 0.00 | 0.17 | 0.00 | -0.30 | 0.40 | 0.05 | 0.09 | 0 |
| | Gradient x Input | 0.17 | 0.00 | 0.17 | 0.00 | 0.34 | 0.61 | 0.05 | 0.09 | 0 |
| | FIS_LR | 0.33 | 0.17 | 0.33 | 0.17 | 0.68 | 0.51 | 0.07 | 0.08 | 0 |
| | FIS_SAEM | **0.67** | 0.17 | 0.17 | 0.17 | **0.81** | **0.78** | 0.07 | 0.08 | 0 |
| | LIME | 0.67 | 0.17 | *0.67* | 0.17 | 0.65 | 0.76 | *0.10* | *0.07* | 3 |
| | SHAP | 0.17 | 0.00 | 0.17 | 0.00 | -0.10 | 0.47 | 0.06 | 0.10 | 0 |
| | Integrated Gradient | *0.83* | 0.00 | 0.00 | 0.00 | *0.86* | *0.84* | 0.09 | 0.08 | 3 |
| | Vanilla Gradient | 0.67 | *0.33* | *0.67* | *0.33* | 0.66 | *0.78* | *0.10* | *0.07* | 5 |
| ANN | SmoothGrad | 0.67 | 0.17 | *0.67* | 0.17 | 0.65 | 0.77 | *0.10* | *0.07* | 3 |
| | Random | 0.17 | 0.00 | 0.17 | 0.00 | -0.30 | 0.40 | 0.05 | 0.11 | 0 |
| | Gradient x Input | 0.33 | 0.00 | 0.33 | 0.00 | 0.34 | 0.62 | 0.06 | 0.10 | 0 |
| | FIS_ANN | 0.33 | 0.00 | 0.33 | 0.00 | 0.57 | 0.23 | 0.08 | 0.08 | 0 |
| | FIS_SAEM | 0.33 | 0.00 | 0.17 | 0.00 | **0.60** | **0.31** | 0.08 | 0.09 | 0 |
| | Decision Trees | 0.33 | 0.00 | 0.17 | 0.00 | 0.56 | 0.22 | 0.06 | 0.09 | - |

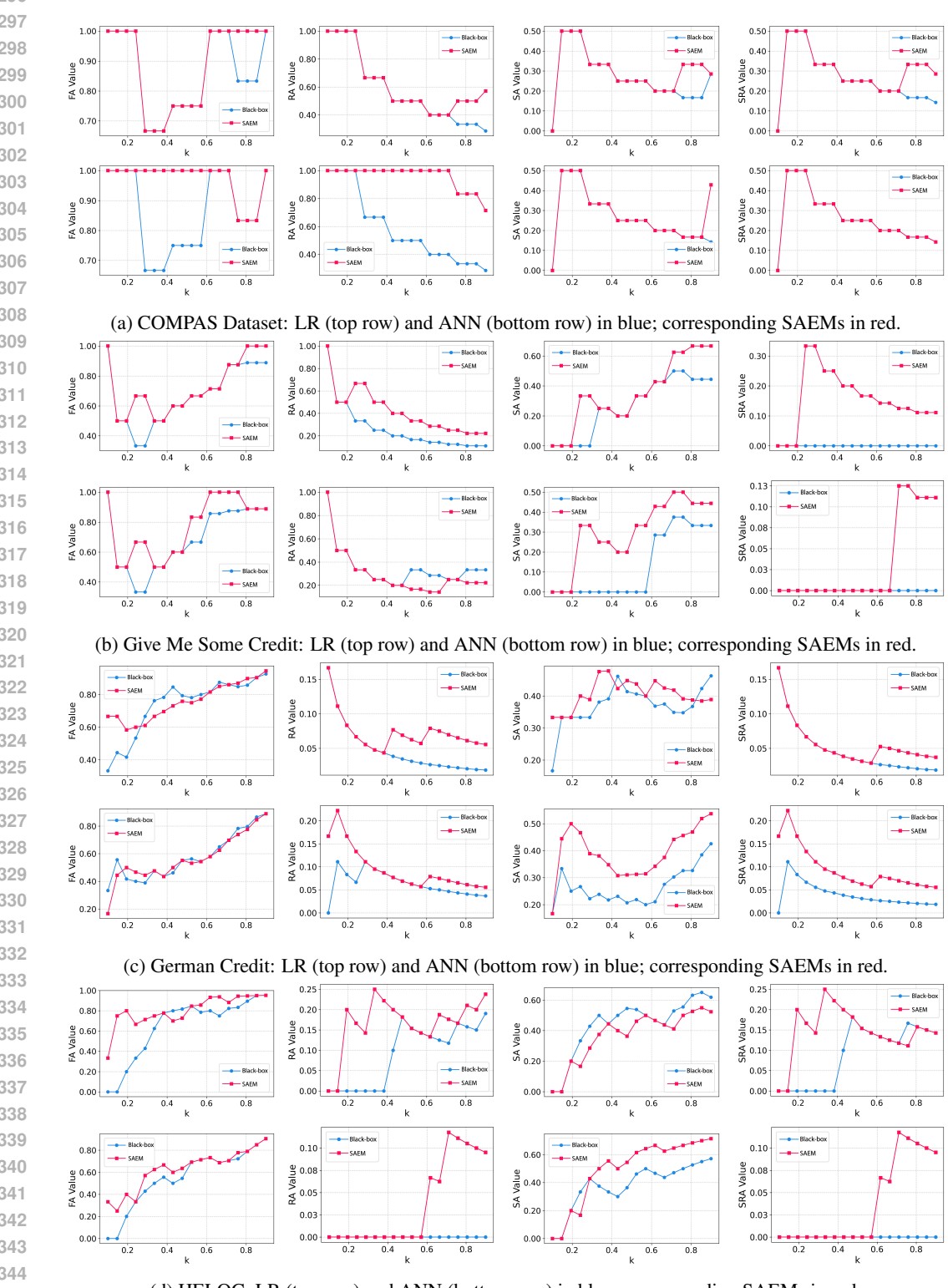

(a) COMPAS Dataset: LR (top row) and ANN (bottom row) in blue; corresponding SAEMs in red.

(b) Give Me Some Credit: LR (top row) and ANN (bottom row) in blue; corresponding SAEMs in red.

(c) German Credit: LR (top row) and ANN (bottom row) in blue; corresponding SAEMs in red.

(d) HELOC: LR (top row) and ANN (bottom row) in blue; corresponding SAEMs in red.

Figure 8: Comparison of faithfulness agreement metrics (FA, RA, SA, SRA) between assumed black box models (LR and ANN) and the identified SAEMs for varying $k$ values on COMPAS, Give Me Some Credit, German Credit and HELOC datasets.

Table 10: Ground-truth and predictive faithfulness results ($k$=0.25) on the Give Me Some Credit dataset for all explanation methods with LR and ANN models. (↑) indicates that higher values are better, and (↓) indicates that lower values are better.

| | Method | FA(↑) | RA(↑) | SA(↑) | SRA(↑) | RC(↑) | PRA(↑) | PGI(↑) | PGU(↓) | #Best |
|---|---|---|---|---|---|---|---|---|---|---|
| | LIME | *1.00* | *1.00* | *1.00* | *1.00* | *1.00* | *1.00* | *0.03* | *0.01* | 8 |
| | SHAP | 0.00 | 0.00 | 0.00 | 0.00 | 0.31 | 0.67 | 0.01 | 0.03 | 0 |
| | Integrated Gradient | *1.00* | *1.00* | *1.00* | *1.00* | *1.00* | *1.00* | *0.03* | *0.01* | 8 |
| | Vanilla Gradient | *1.00* | *1.00* | *1.00* | *1.00* | *1.00* | *1.00* | *0.03* | *0.01* | 8 |
| LR | SmoothGrad | *1.00* | *1.00* | *1.00* | *1.00* | *1.00* | *1.00* | *0.03* | *0.01* | 8 |
| | Random | 0.33 | 0.33 | 0.00 | 0.00 | -0.24 | 0.42 | 0.01 | 0.03 | 0 |
| | Gradient x Input | 0.67 | 0.00 | 0.67 | 0.00 | 0.45 | 0.73 | 0.02 | *0.01* | 1 |
| | FIS_LR | 0.33 | 0.00 | 0.00 | 0.00 | 0.67 | 0.42 | *0.03* | *0.01* | 2 |
| | FIS_SAEM | **0.66** | **0.66** | 0.00 | 0.00 | **0.76** | **0.56** | *0.03* | *0.01* | 2 |
| | LIME | *1.00* | *1.00* | *1.00* | *1.00* | 0.94 | 0.91 | *0.11* | *0.01* | 6 |
| | SHAP | 0.33 | 0.33 | 0.33 | 0.33 | 0.36 | 0.67 | 0.10 | 0.02 | 0 |
| | Integrated Gradient | 0.67 | 0.67 | 0.67 | 0.67 | 0.98 | 0.96 | *0.11* | *0.01* | 2 |
| | Vanilla Gradient | *1.00* | *1.00* | *1.00* | *1.00* | *0.99* | *0.98* | *0.11* | *0.01* | 8 |
| ANN | SmoothGrad | *1.00* | *1.00* | *1.00* | *1.00* | 0.94 | 0.91 | *0.11* | *0.01* | 6 |
| | Random | 0.33 | 0.33 | 0.00 | 0.00 | -0.24 | 0.42 | 0.02 | 0.11 | 0 |
| | Gradient x Input | 0.33 | 0.33 | 0.33 | 0.33 | 0.26 | 0.60 | 0.10 | 0.02 | 0 |
| | FIS_ANN | 0.33 | 0.33 | 0.00 | 0.00 | 0.62 | 0.32 | 0.10 | 0.02 | 0 |
| | FIS_SAEM | **0.67** | 0.33 | **0.33** | **0.33** | **0.80** | **0.73** | ***0.11*** | ***0.01*** | 2 |
| | Decision Trees | 0.67 | 0.00 | 0.00 | 0.00 | 0.82 | 0.79 | 0.02 | 0.01 | - |

