# OpenReview forum: "EXAGREE: Towards Explanation Agreement in Explainable Machine Learning"
_ICLR.cc/2025/Conference — Submitted to ICLR 2025_

### Official Review · Reviewer_rmJR · 2024-10-28

**Soundness:** 1
**Presentation:** 1
**Contribution:** 1
**Rating:** 1
**Confidence:** 4

**Summary:**

The paper proposes to address the problem of explanation disagreement across different models (of same performance on a dataset, called a Rashomon set), and across different explanation techniques. The authors say that disagreement between explanations hurts reliability of the model if and when deployed in high-stakes environment. They propose EXAGREE where they talk about 4 avenues of such disagreement, and propose an approach that **apparently** addresses the problem.

**Strengths:**

The problem that the paper focuses on is interesting and the paper does a good job at disentangling the four scenarios of explanation disagreement.

Originality and Significance (of the problem): High
Quality and Clarity: Poor

**Weaknesses:**

The paper has a lot of weakness in my opinion and I delineate them as follows. You do not need to address all of them in the rebuttal, I have marked points that are major weakness, you can focus on them.

1. Starting with Figure 1 on Page 1 -- totally unclear. How does the right side of the figure indicate agreement while the left does not? (this figure is not crucial to understand the paper, but I am just making a point of why the paper is so unclearly written and you do not need to address this point in the rebuttal, there are many more important problems. )

2. [Major] In Section 2 (Preliminaries), there are several mistakes in the formulation:

**a)** M* is not defined in Eq. 1. I assume what you mean is the optimal model by M* (the model with the best performance on the data).

**b)** a_1, a_2,...,a_p are not defined as features.

**c)** In Equation 1, I think the sign should be >= instead of <= (the loss of the other models will be higher than M* (assuming M* refers to optimal model)

**d)** Why is $\epsilon$ multiplied by L(M*), usually similar performing models are supposed to be an absolute threshold, not relative to the loss of the original model.

3. [Most Major] In Section 2.1, you mention 4 axes of explanation disagreement. I am with you on the model and method disagreement, but what does stakeholder and ground truth disagreement mean?

**a)** In Stakeholder disagreement you mention "Different stakeholders in S might prefer different rankings". Why would that ever happen? Do these stakeholders want something that is not real and want fake explanations that aligns with their mental expectations of what a model should care about? What is an example of such stakeholders wanting something different from reality? Your current example of a data scientist wanting statistical significance and domain expert valuing different features **does not** answer this. Does wanting statistical significance and valuing different features amount to wanting different ranking of features (this also assumes that these rankings won't be same in the first place)? And even if they want different rankings, should a technique be optimized to suit their demands and provide fake explanations?

**b)** In ground truth disagreement you mention "ground truth interpretations (i think you mean explanations) from interpretable models can conflict with post-hoc explanations" -- yeah this is obvious, but does it matter? If I have an interpretable model, why do I care what a post-hoc technique says, I will never ever use that for such a model? Why would one do that?

**c)** In the primary objective of the paper you write: "identify a well-performing model that minimizes disagreement (or maximizes agreement) between model explanations and stakeholder expectations. " -- where stakeholder expectation is either something we should not give them because it is fake or something I do not understand. Your rebuttal will help me understand if it is the latter (a solid convincing example is required).

5. Section 2.3, I think the title of that subsection should be "Evaluation Metrics" and not "Evaluation Matrices". Anyway, in that section you mention two metrics: faithfulness assessment and fairness assessment. In faithful assessment you give a further breakdown in several metrics -- however, most of these metrics seems highly correlated, for e.g., the attribution values directly affect the ranking, so FA, RA, RC, SA, SRA, PRA should have extreme correlation (when you consider the attribution values with its sign). Can you measure this please? If they are highly correlated, then that is effectively one metric and not six. If they are not correlated, can you explain why that is not the case (except the case where you consider the absolute values, its obvious then).

6. You have used the word "mask" in line 256, without defining, what does it mean? Also in line 259 what does the phrase "bridging the gap between models in Rashomon set and their feature attributions mean" mean? each model in the Rashomon set (or any model for that matter) has a feature attribution (produced by any technique), what is the gap between them -- they don't even lie in the same space, one is in the weight space and other is in the explanation space (they have different dimensionality).

7. [Major] You have this complicated (and not at all well explained) pipeline of computing SAEM which is basically the models that have the highest agreement with stakeholder requirements. Please tell me why do you need this pipeline? You have the models that are trained on that dataset, you can use that to compute the set of the models that are best performing (according to some threshold $\epsilon$), and then compute the explanations of these models using whatever method you like, and then just give those models to the stakeholders with whom they have the highest agreement. What is the job of the models like DMAN, you are just using that to predict the model that will have the highest agreement (is that a correct understanding of the model's job?), but you don't need that you have the ground truth and you even say this in lines 268-269.

8. [Major] The above were the problems with the problem setup, now coming to the experiment section.
**a)** In table 1 you mention k= 0.25, what does k stand for? You have used k and l in Section 2.1 without ever defining them. For the same reason, I do not comprehend what does the k in caption of Figure 3 stand for.

**b)** Why is SAEM technique bolded in Tables 1 and 2. The standard practice in ML papers to the bold the best performing method, but your technique, and this is misleading.

**c)** If SAEM selects the models with the highest agreement with the stakeholder, why is it not the best for LR in Table 1 and 2? (especially in Table 2, it is pretty behind techniques that are not optimized for agreement, which alludes to the case that explanation disagreement might not even show up in actual experiments?)

**d**) how is FIS_LR computed? Is this not just the feature importance and since you are using ground-truth explanations from pre-trained LR, this should have 100% agreement and should be the best 8 number of times (currently in Table 1 it is best 0 times)?

**Questions:**

I have listed all the questions in the weakness section, the authors can focus on the major problems if they like.

---

> ### Author Response · Authors · 2024-11-15
>
> We thank reviewer for recognition of the work as interesting. We clarify several points that may have led to misunderstandings below. Our goal with EXAGREE is not to present a comprehensive solution to explanation disagreement. Rather, we aim to provide a framework that begins to **reduce** explanation disagreement. We answer your major concerns first.
>
> **Major**
> ---
> >  [Major] In Section 2 (Preliminaries), there are several mistakes in the formulation:
> 	**a)**  M* is not defined in Eq. 1. I assume what you mean is the optimal model by M* (the model with the best performance on the data).
> 	**b)**  a_1, a_2,...,a_p are not defined as features.
> 	**c)**  In Equation 1, I think the sign should be >= instead of <= (the loss of the other models will be higher than M* (assuming M* refers to optimal model)
> 	**d)**  Why is  ϵ  multiplied by L(M*), usually similar performing models are supposed to be an absolute threshold, not relative to the loss of the original model.
>
> We respectfully disagree with some of the points raised and appreciate the opportunity to clarify:
> -   **a)** We apologize for this oversight and It indeed represents the optimal model.
> -   **b)** The variables $a_1, a_2, ...a_p$​ represent feature attributions, **not** the features themselves.
> -   **c)** The sign in Equation 1, **≤**, is **correct**. It indicates that all models in the Rashomon set perform within a specified threshold relative to $M^{*}$.
> -   **d)** The threshold $\epsilon$ can indeed be defined as either a relative or absolute value. Both approaches are valid, and more details can be found in cited works.
>
>
> 3.  [Most Major] In Section 2.1, you mention 4 axes of explanation disagreement. I am with you on the model and method disagreement, but what does stakeholder and ground truth disagreement mean?
>
> > **a)**  In Stakeholder disagreement you mention "Different stakeholders in S might prefer different rankings". Why would that ever happen? Do these stakeholders want something that is not real and want fake explanations that aligns with their mental expectations of what a model should care about? What is an example of such stakeholders wanting something different from reality? Your current example of a data scientist wanting statistical significance and domain expert valuing different features  **does not**  answer this. Does wanting statistical significance and valuing different features amount to wanting different ranking of features (this also assumes that these rankings won't be same in the first place)? And even if they want different rankings, should a technique be optimized to suit their demands and provide fake explanations?
>
> Response:
> Thank you for raising this point. The stakeholder disagreement emerges not because stakeholders want "fake" explanations, but because different roles require different aspects for their specific tasks, while respecting medical constraints.
>
> Consider a more concrete breast cancer diagnostic model in a healthcare setting, with real-world diagnostic features, such as **mammogram findings** (e.g., microcalcifications, mass shape/margins), **ultrasound results** (e.g., solid vs. cystic masses), **biopsy cell characteristics** (e.g., nuclear grade, mitotic count), **tumor size**, **age at diagnosis**, **family history**, and **breast density**. The **model developer** considers all features to optimize performance, while other stakeholders need explanations that emphasize features relevant to their decision-making stage:
>
> 1.  **Radiologists** primarily use **imaging features** during initial screenings. If the model's explanation emphasizes biopsy results (which aren't available during screening), this creates an operational mismatch - not because the biopsy features aren't important, but because radiologists need to understand the model's reasoning based on imaging data available at their decision point.
>
> 2.  **Pathologists** focus on **cellular characteristics** from biopsies, such as nuclear grade and mitotic count, which are essential for assessing malignancy at the cellular level. Conflicts arise if the model emphasizes imaging features over biopsy data, as imaging cannot provide the cellular detail pathologists need.
>
> 3.  **Primary Care Physicians** make initial referral decisions using **family history, age, and previous records**. They need explanations focused on these early-stage risk indicators rather than specialized imaging or biopsy findings.
>
> These stakeholders aren't seeking incorrect explanations - they need explanations that reflect how the model uses features accessible for their roles, making the model more actionable and useful. This stakeholder disagreement represents just one of the four fundamental sources and the other three sources all contribute to the broader challenge in practice. We will explain the ground truth disagreement below.

---

> > ### Author Response · Authors · 2024-11-15
> >
> > > **b)**  In ground truth disagreement you mention "ground truth interpretations (i think you mean explanations) from interpretable models can conflict with post-hoc explanations" -- yeah this is obvious, but does it matter? If I have an interpretable model, why do I care what a post-hoc technique says, I will never ever use that for such a model? Why would one do that?
> >
> > Response:
> > While it might seem unnecessary to apply post-hoc techniques to interpretable models, this source is critical for understanding explanation disagreement.
> >
> > Consider the healthcare scenario again, where the decision tree's interpretation could be problematic despite high accuracy: a decision tree flags small cell size (<4mm) more important than larger cell sizes (>4mm), **contradicting** established medical knowledge where larger cell normally indicates positive prediction. Even though the tree is "interpretable," its decision path violates known biological mechanisms. Interestingly, post-hoc methods (e.g., SHAP) might rank features differently on the same tree, rating larger cell sizes as more important, aligning with medical knowledge.
> >
> > This disagreement between the tree's structure and SHAP's analysis raises an interesting question: Is SHAP wrong because it contradicts the "interpretable" tree's structure, or the algorithm itself untrusted? Or does this disagreement reveal problems with how the tree learned to make predictions?
> > We don't consider either explanation wrong, but the conflict is real which can be indicated by metrics. That is why studying these disagreements needs more attention.
> >
> > **c)**  In the primary objective of the paper you write: "identify a well-performing model that minimizes disagreement (or maximizes agreement) between model explanations and stakeholder expectations. " -- where stakeholder expectation is either something we should not give them because it is fake or something I do not understand. Your rebuttal will help me understand if it is the latter (a solid convincing example is required).
> >
> > Response: We hope the examples above help clarify the role of stakeholder-specific expectations in explanation disagreements. Thank you for the opportunity to clarify.
> >
> >
> >
> > >  [Major] You have this complicated (and not at all well explained) pipeline of computing SAEM which is basically the models that have the highest agreement with stakeholder requirements. Please tell me why do you need this pipeline? You have the models that are trained on that dataset, you can use that to compute the set of the models that are best performing (according to some threshold  ϵ), and then compute the explanations of these models using whatever method you like, and then just give those models to the stakeholders with whom they have the highest agreement. What is the job of the models like DMAN, you are just using that to predict the model that will have the highest agreement (is that a correct understanding of the model's job?), but you don't need that you have the ground truth and you even say this in lines 268-269.
> > >  You have used the word "mask" in line 256, without defining, what does it mean? Also in line 259 what does the phrase "bridging the gap between models in Rashomon set and their feature attributions mean" mean? each model in the Rashomon set (or any model for that matter) has a feature attribution (produced by any technique), what is the gap between them -- they don't even lie in the same space, one is in the weight space and other is in the explanation space (they have different dimensionality).
> >
> >
> > You are correct that, in theory, one could compute explanations for all models in the Rashomon set and select those with the highest stakeholder agreement. However, the challenge lies in the fact that it is not feasible to sample all models from the Rashomon set. Current methods can only approximate this set, so directly exploring the relationship between models and their feature attributions becomes a more practical approach.
> >
> > DMAN addresses this challenge by efficiently approximating the non-linear mapping between model space (the weight space) and attribution space (the explanation space). As you correctly noted, these spaces have different dimensionalities, especially the models in the model space. The "mask" (line 256) refers to the characterization of models in the Rashomon set, enabling DMAN to learn this complex relationship between spaces.
> >
> > Through this approach, DMAN enables us to **predict attributions** for new models without directly computing them. More importantly, we can optimize models with the desired attribution patterns. Regarding lines 268-269: Yes, since DMAN provides approximations, we need to evaluate the final results.

---

> ### Author Response · Authors · 2024-11-15
>
> >  [Major] The above were the problems with the problem setup, now coming to the experiment section.
> 	**a)**  In table 1 you mention k= 0.25, what does k stand for? You have used k and l in Section 2.1 without ever defining them. For the same reason, I do not comprehend what does the k in caption of Figure 3 stand for.
>     **b)**  Why is SAEM technique bolded in Tables 1 and 2. The standard practice in ML papers to the bold the best performing method, but your technique, and this is misleading.
> 	**c)**  If SAEM selects the models with the highest agreement with the stakeholder, why is it not the best for LR in Table 1 and 2? (especially in Table 2, it is pretty behind techniques that are not optimized for agreement, which alludes to the case that explanation disagreement might not even show up in actual experiments?)
> 	**d**) how is FIS_LR computed? Is this not just the feature importance and since you are using ground-truth explanations from pre-trained LR, this should have 100% agreement and should be the best 8 number of times (currently in Table 1 it is best 0 times)?
>
> a) We apologize for this oversight. $k$ represents the top percentage of features considered. For example, with $k=0.25$ in a list of 12 features, we consider the top 3 most important features.
>
> b) and c): As the first framework specifically targeting explanation agreement improvement, our experimental setup focuses on demonstrating relative improvements. This is why we bolded the improved metrics. This does not mean our method is always the best performing method compared to any explanation method in explaining any model.
>
> We acknowledge that our framework **cannot** satisfy all metrics for all stakeholders due to performance constraints. As formally proven in Section D and Appendix "Constrained Sorting and Ranking in a Rashomon Set".
> Regarding Table 2, gradient-based methods achieve perfect agreement for LR models because they directly compute derivatives that match ground truth coefficients. However, these same methods show significantly lower agreement when explaining ANN models. This highlights a key limitation - while existing methods may perform well for specific model types, we provide more consistent agreement improvements across different model architectures and stakeholder needs.
>
> d) FIS_LR is computed based on the loss change, described in cited works. The definition is referred as $\varphi_{i}(\textbf{X}, {y})$ = $L_{emp}$($f(\{X\}\_{\setminus i}, \textbf{y})$ - $L_{emp}(f(\textbf{X}), \textbf{y})$ for feature importance of feature $i$, where $\textbf{X}_{\setminus i}$ is the input when the feature of interest is replaced by an independent variable.
>
> As acknowledged in Section 3.1, this is a model-agnostic post-hoc explanation method that doesn't guarantee perfect agreement with ground truth. This is similar to other established methods like SHAP, which shows 50% agreement in our experiments but the method remains widely used and trusted in practice.
>
> ---
>
> **Minor** Questions
>
> >  Starting with Figure 1 on Page 1 -- totally unclear. How does the right side of the figure indicate agreement while the left does not? (this figure is not crucial to understand the paper, but I am just making a point of why the paper is so unclearly written and you do not need to address this point in the rebuttal, there are many more important problems. )
>
> We will either improve or remove this figure.
>
>
> >  Section 2.3, I think the title of that subsection should be "Evaluation Metrics" and not "Evaluation Matrices". Anyway, in that section you mention two metrics: faithfulness assessment and fairness assessment. In faithful assessment you give a further breakdown in several metrics -- however, most of these metrics seems highly correlated, for e.g., the attribution values directly affect the ranking, so FA, RA, RC, SA, SRA, PRA should have extreme correlation (when you consider the attribution values with its sign). Can you measure this please? If they are highly correlated, then that is effectively one metric and not six. If they are not correlated, can you explain why that is not the case (except the case where you consider the absolute values, its obvious then).
>
> Thank you for your suggestion and the title will be corrected.
>
> We agree that FA, RA, RC, SA, SRA, PRA are correlated. It's important to note that we did not define these faithfulness metrics but followed the work (OpenXAI [https://arxiv.org/pdf/2206.11104](https://arxiv.org/pdf/2206.11104)).
>
> ---
>
> Thank you for all your feedback.

---

> ### Comment · Reviewer_rmJR · 2024-11-27
> **Reviewer Response to Author Rebuttal**
>
> I appreciate the time taken by the authors to read and give a response to my review. I apologize for the lateness in my response. I have follow-up questions that I am listing:
> >  1. In Equation 1, I think the sign should be >= instead of <= (the loss of the other models will be higher than M* (assuming M* refers to optimal model)
> > Authors said: The sign in Equation 1, ≤, is correct. It indicates that all models in the Rashomon set perform within a specified threshold relative to M*.
>
> If M* is the optimal model, then how can the loss of any other model be lower than that and therefore I thought the sign should be reversed. Please correct me if I am wrong.
>
> > 2. Authors said: "These stakeholders aren't seeking incorrect explanations - they need explanations that reflect how the model uses features accessible for their roles, making the model more actionable and useful."
>
> The example you give is still not convincing to me. There are radiologists, pathologists, and PCPs in this setup and they want explanations from the model which are accessible for their roles, I can understand that. There are a few options they can use:
> a) build separate models using the features that are accessible to each of the people. Now I know, you might think this will give a low-performing model than a model trained on all features, in which case:
> b) just get the explanations from the model trained on features, rank them based on importance and show the relevant groups of features accessible to each person.
>
> Is this not an easier and effective solution? If not, then please let me know why.
>
> > 3. Authors said: Consider the healthcare scenario again, where the decision tree's interpretation could be problematic despite high accuracy: a decision tree flags small cell size (<4mm) more important than larger cell sizes (>4mm), contradicting established medical knowledge where larger cell normally indicates positive prediction.
>
> If such a situation arises, we don't use such a model, because we know it is **wrong** and dangerous to use such a model. Since it is an interpretable model, we know exactly how the model is making the decision. And if it is totally opposite of how biology works, we should not use such a model. (If you disagree, please give me an argument about why we should use continue using such a model, just because its accurate on a dataset the model was trained on or a related test dataset?)  Therefore I don't see any argument for ground-truth disagreement.
>
> > 4. We hope the examples above help clarify the role of stakeholder-specific expectations in explanation disagreements. Thank you for the opportunity to clarify.
>
> Not yet unfortunately.
>
> > 5. The "mask" (line 256) refers to the characterization of models in the Rashomon set
>
> What does this line mean? Please use the terms that are defined in the paper. After reading your rebuttal it was much more clear what the job of DMAN is, but because of missing definitions when a reviewer is reading your paper it completely derails them. The same thing happened in the experimental section with missing "k". Also, you only answered my question about DMAN, please also explain the role of SAEM, Multi-heads architecture in Section 3.1.
>
> > 6. As the first framework specifically targeting explanation agreement improvement, our experimental setup focuses on demonstrating relative improvements. This is why we bolded the improved metrics.
>
> No, you bolded the row for your technique, not the "improved metrics" -- in which case you would have bolded the best performing technique per column.

---

> > ### Author Response · Authors · 2024-11-27
> >
> > >I appreciate the time taken by the authors to read and give a response to my review. I apologize for the lateness in my response. I have follow-up questions that I am listing:
> >
> > Response: We thank the reviewer for the follow-up questions. This conversation provides us with another valuable perspective, enabling us to refine and improve our study in the future.
> >
> > > If M* is the optimal model, then how can the loss of any other model be lower than that and therefore I thought the sign should be reversed. Please correct me if I am wrong.
> >
> > Response: : Equation 1 indeed uses the definition of the Rashomon set, where $\epsilon$ serves as a performance boundary. While $M^*$ represents the optimal model, $L(M)$ can be lower than $L(M^*)$ due to it being scaled by (1+$\epsilon$). This condition ensures models in the set perform comparably within the specified threshold.
> >
> > > The example you give is still not convincing to me. There are radiologists, pathologists, and PCPs in this setup and they want explanations from the model which are accessible for their roles, I can understand that. There are a few options they can use: a) build separate models using the features that are accessible to each of the people. Now I know, you might think this will give a low-performing model than a model trained on all features, in which case: b) just get the explanations from the model trained on features, rank them based on importance and show the relevant groups of features accessible to each person. Is this not an easier and effective solution? If not, then please let me know why.
> >
> > Response: That is an excellent point, and it highlights the practical challenges we intented to address in this work.
> > While your suggestion is both reasonable and feasible, the explanation disagreement problem is more complex, as disagreements can arise from other sources beyond feature accessibility. For instance, let’s set aside performance considerations and assume a stakeholder uses a black-box model with SHAP (or even an interpretable model that is misinterpreted). If the explanation generated contradicts the stakeholder's domain knowledge, as you noted in your next question, the model becomes unusable. In such cases, the question becomes: what steps should the stakeholder take next, or how can we identify a model that aligns with their expectations while remaining actionable? This underscores the importance of systematically considering explanation disagreements and the need to find models that better serve diverse stakeholder needs.
> >
> > > If such a situation arises, we don't use such a model, because we know it is  **wrong**  and dangerous to use such a model. Since it is an interpretable model, we know exactly how the model is making the decision. And if it is totally opposite of how biology works, we should not use such a model. (If you disagree, please give me an argument about why we should use continue using such a model, just because its accurate on a dataset the model was trained on or a related test dataset?) Therefore I don't see any argument for ground-truth disagreement.
> >
> > Response: We fully agree that such a **wrong** model should not be used. Ground-truth disagreement is not something we see as problematic but rather as a source of explanation disagreement. Our focus is to (1) identify such sources and (2) provide a systematic approach to avoid relying on such incorrect models.
> >
> > > Not yet unfortunately.
> >
> > Response: We are sorry to hear that and thank you for raising these questions. We hope the explanations provided above clarify some of the points. If not, please feel free to reply before the deadline, as we may not be able to respond afterward. Once again, we appreciate your concerns and suggestions, as we believe such discussions are the **true** purpose of the review and rebuttal process.
> >
> > > What does this line mean? Please use the terms that are defined in the paper. After reading your rebuttal it was much more clear what the job of DMAN is, but because of missing definitions when a reviewer is reading your paper it completely derails them. The same thing happened in the experimental section with missing "k". Also, you only answered my question about DMAN, please also explain the role of SAEM, Multi-heads architecture in Section 3.1.
> >
> > Response: SAEM refers to the model we identify from the Rashomon set that aligns its explanations with stakeholder expectations. The multi-head architecture is designed to address diverse stakeholder needs by generating multiple possible solutions.
> >
> >
> > > No, you bolded the row for your technique, not the "improved metrics" -- in which case you would have bolded the best performing technique per column.
> >
> > Response: Thank you for your suggestion. The improved metrics refer to measures such as FA and RA, evaluated within the same model and explanation methods. We will refine our evaluation approach, particularly to ensure clarity and accuracy in presenting results.

---

### Official Review · Reviewer_9c6d · 2024-11-03

**Soundness:** 3
**Presentation:** 1
**Contribution:** 3
**Rating:** 3
**Confidence:** 4

**Summary:**

The paper begins with an implicit premise that there exists an ideal model explanation for a given set of model predictions. Given this kind of explanation obtained from an oracle, the paper presents an algorithm to select a model from a Rashomon set of models that matches the expected explanations. The paper terms this as an SAEM (stakeholder aligned explanation model). "Explanation model" here does not seem to refer to an explanation method, but a predictive model (one picked from the existing Rashomon set) that best matches the explanations desired by a stakeholder.

**Strengths:**

The scientific contribution presented in the paper is valid and interesting. The proposed method has the means to create machine learning models that can produce explanations that match an arbitrary ideal. The paper presents a new, model-agnostic method to accomplish this, which would be a meaningful and important contribution to the field.

**Weaknesses:**

The paper’s formalization of four types of disagreements lacks sufficient motivation and clarity, especially in comparison to the foundational work it draws from. For example, the classification of "model disagreement" as a type of "explanation disagreement" is unclear - different models that produce the same predictions can indeed have different internal mechanisms of doing so - in this case explanations should disagree and illuminate this fact rather than obscure it. This notion aligns with the concept of Rashomon sets, where multiple models with similar predictive performance can have significantly different decision boundaries. I encourage the authors to clarify their rationale for categorizing model disagreement within the explanation disagreement framework and to elaborate on how their approach handles cases where differing explanations for similar predictions might reveal essential model behaviors rather than obscure them. There is no "model disagreement" problem.

The paper's contributions then are better studied and understood in the context of a related line of inquiry about adversarial attacks and explanation fairwashing, such as Slack et al (https://doi.org/10.1145/3375627.3375830). The primary problem addressed appears to involve modifying the explanations produced by a Rashomon set of models to align with a predefined set of explanations from an external oracle. This connection is effectively illustrated, though not explicitly addressed, in Table 1, where metrics such as FA, RA, SRA, and others are reported for the ANN model. Notably, the original OpenXAI benchmark (Agarwal et al.: https://dl.acm.org/doi/10.5555/3600270.3601418) does not provide ground truth for ANNs. What this paper does (I think) is use the LR coefficients as ground-truths to measure metrics such as FA, RA, SRA, etc against a **different** model - an ANN! I recommend the authors clarify their novel approach to calculating these metrics for the ANN model and discuss the ethical implications of aligning ANN explanations with LR model coefficients.

Finally, the paper’s entire framing around "explanation agreement" (motivated in Section 2) could be made clearer. Rather than resolving "model disagreement," the proposed SAEM approach seems to modify model explanations without altering predictive accuracy, which could be viewed as a form of adversarial attack on explanations. I encourage the authors to address how this approach contrasts with adversarial manipulations of explanations (if at all), discuss potential connections to explanation fairwashing, and consider any ethical implications that arise from intentionally adjusting explanations while maintaining predictive outputs. What is presented in this paper as an SAEM to resolve the apparent "model disagreement" class of explanation disagreement problems is essentially a means to make the FIS score for the ANN model to match the coefficients from the LR model trained on the same data - this is an adversarial attack.

**Questions:**

It would be good for the paper to define terms such as "explanation model" before using them (it is possible that I've misunderstood this term in my review). There seem to be other consistency errors that could elevate the writing and clarity. For example, Table 1 has a model class and explanation method in columns 1 and 2, under the common heading method. However the last row has "decision trees" listed as an explanation method (my understanding is this should be a model - in which case what is the explanation method used?).

The figures 1 and 2 were similarly unclear to me - the symbols used were not explained (what do the curved arrows represent? what is the lightbulb? what do the question-mark and check-mark mean respectively?) and did not help with my understanding. These could be omitted entirely in my opinion without any impact on the papers clarity.

Most critically though, perhaps the paper needs to dispel the notion of promoting a nebulous notion of "explanation agreement", as motivated in section 2, and recognise the algorithm for what it does - produce models that can maintain predictive accuracy while generating explanations that can match different ones. The paper is thus not resolving "model disagreement", but introducing an adversarial attack that maintains predictions and modifies explanations.

To reiterate, I think the method is good and the contributions are interesting and valuable, but I think the framing of "explanation agreement" can be replaced with a "adversarial manipulations of explanations" to clarify the writing.

---

> ### Author Response · Authors · 2024-11-14
>
> >The paper’s formalization of four types of disagreements lacks sufficient motivation and clarity, especially in comparison to the foundational work it draws from. For example, the classification of "model disagreement" as a type of "explanation disagreement" is unclear - different models that produce the same predictions can indeed have different internal mechanisms of doing so - in this case explanations should disagree and illuminate this fact rather than obscure it... There is no "model disagreement" problem.
>
> Response: We appreciate the reviewer’s feedback regarding the motivation behind the different types of disagreement in our paper. We would like to clarify the following points:
>
> "Model disagreement" is **not** a problem; rather, it represents a source of potentially conflicting explanations, consistent with the Rashomon set concept, which in turn leads to explanation disagreement. Similarly, this perspective applies across all four types we present in the paper. Explanation disagreement becomes problematic when these differences are unrecognized and explanations are trusted blindly, like in the criminal justice case.
>
> We are with the reviewer on this perspective and indeed, we take advantage of model disagreement as a means to reduce explanation disagreement, where our aim is to identify a model from a Rashomon set that provides personalized explanations based on their specific needs and preferences.
>
> > The paper's contributions then are better studied and understood in the context of a related line of inquiry about adversarial attacks and explanation fairwashing, such as Slack et al. The primary problem addressed appears to involve modifying the explanations produced by a Rashomon set of models to align with a predefined set of explanations from an external oracle.
>
> Response: We sincerely thank the reviewer for recognizing the experimental design and for your insightful understanding of the use of LR coefficients as ground truth for the ANN model. This was indeed the approach we intended, and we appreciate the clarity of your observation.
>
> We agree that our work shares conceptual space with research on adversarial attacks and explanation fairwashing. The intersection between adversarial attacks and the Rashomon set concept is an interesting direction for further exploration.
>
> > Finally, the paper’s entire framing around "explanation agreement" could be made clearer. Rather than resolving "model disagreement," the proposed SAEM approach seems to modify model explanations without altering predictive accuracy, which could be viewed as a form of adversarial attack on explanations. I encourage the authors to address how this approach contrasts with adversarial manipulations of explanations (if at all), discuss potential connections to explanation fairwashing, and consider any ethical implications that arise from intentionally adjusting explanations while maintaining predictive outputs. What is presented in this paper as an SAEM to resolve the apparent "model disagreement" class of explanation disagreement problems is essentially a means to make the FIS score for the ANN model to match the coefficients from the LR model trained on the same data - this is an adversarial attack.
>
> Response: We agree that there are conceptual similarities between our approach and adversarial attacks, particularly in the idea of modifying model explanations. However, the objectives, methods, and context of our work are distinct.
>
> Unlike adversarial attacks designed to exploit vulnerabilities in explanation methods, our aim is to systematically explore and improve model explanations to better align with stakeholder needs. Certainly, we believe that similar ideas from adversarial attack research can be applied in this context to address explanation disagreement.
>
> While adversarial attacks typically operate by perturbing inputs to manipulate explanations while maintaining potentially biased behavior, our method is built on the Rashomon set framework. We're not modifying/fooling a single model's explanations, but rather exploring the model space and identifying models that provide desired explanations. It is noted that a Rashomon set can contain models of different architectures, e.g., a tree or a network, not just wrapper modifications. Our mask-based sampling method represents one approach to exploring the space.
> Interestingly, scaffolded models produced by adversarial attacks could theoretically exist within the Rashomon set if they meet the performance condition. Further exploration of this intersection could be a valuable direction, though it is outside the scope of our current work.
>
> Finally, while we acknowledge that any technique modifying explanations could potentially be misused, EXAGREE's optimization within the Rashomon set represents a different perspective which helps users understand the range of feature attributions (attribution sets) instead of a single ranking.

---

> > ### Author Response · Authors · 2024-11-14
> >
> > **Questions:**
> >
> > > It would be good for the paper to define terms such as "explanation model" before using them (it is possible that I've misunderstood this term in my review). There seem to be other consistency errors that could elevate the writing and clarity. For example, Table 1 has a model class and explanation method in columns 1 and 2, under the common heading method. However the last row has "decision trees" listed as an explanation method (my understanding is this should be a model - in which case what is the explanation method used?).
> >
> > You are correct that decision trees should be listed as a model class rather than an explanation method. Decision trees should ideally be placed across two cells in the “models” and “methods” columns, as they offer built-in interpretability through Gini importance.  We will revise Table 1 to reflect this distinction clearly.
> >
> > Your understanding of StakeholderAligned Explanation Models is also accurate, which refers to a predictive model that produces stakeholder desired explanations.
> >
> > > The figures 1 and 2 were similarly unclear to me - the symbols used were not explained (what do the curved arrows represent? what is the lightbulb? what do the question-mark and check-mark mean respectively?) and did not help with my understanding. These could be omitted entirely in my opinion without any impact on the papers clarity.
> >
> > Regarding Figures 1 and 2, we appreciate the feedback. The curved arrows, lightbulb, question mark, and check mark were intended as visual aids to represent explanation disagreement problem and potential solver. We will consider either removing Figure 1 or remaking it, and we will revise Figure 2 to better illustrate the core framework.
> >
> > > Most critically though, perhaps the paper needs to dispel the notion of promoting a nebulous notion of "explanation agreement", as motivated in section 2, and recognise the algorithm for what it does - produce models that can maintain predictive accuracy while generating explanations that can match different ones. The paper is thus not resolving "model disagreement", but introducing an adversarial attack that maintains predictions and modifies explanations.
> >
> > We appreciate the reviewer's thoughtful critique of our framing. We view explanation disagreement as an important emerging research direction in explainable AI. Our intent was not to imply a complete solution to "explanation agreement," but rather to propose a framework that advances initial understanding of and methods for **reducing** explanation disagreement.
> >
> > > To reiterate, I think the method is good and the contributions are interesting and valuable, but I think the framing of "explanation agreement" can be replaced with a "adversarial manipulations of explanations" to clarify the writing.
> >
> > We appreciate the reviewer's recognition of our method's technical merit and their suggestion regarding framing through above discussion. While we acknowledge the methodological similarities with adversarial manipulations and will expand our discussion of these connections, we believe the explanation agreement serves an important purpose. This relatively new research direction, while challenging to study and accept, opens important questions about human-centered model interpretability that extend beyond the scope of adversarial manipulation. While adversarial manipulation can offer valuable insights, we believe properly establishing the broader context of explanation agreement should come first.

---

### Official Review · Reviewer_ryTs · 2024-11-04

**Soundness:** 1
**Presentation:** 2
**Contribution:** 1
**Rating:** 1
**Confidence:** 4

**Summary:**

This paper addresses the problem of explanation disagreement (or explanatory multiplicity), where (post-hoc) explanations of a given machine learning model conflicts with one another. The authors propose a new framework called EXAGREE to find a model, stakeholder-aligned explanation model (SAEM), that provides explanations (feature attribution rankings) in accordance with stakeholder desiderata.

**Strengths:**

The technical framework of end-to-end optimization problem which involves constructing the Rashomon set, DMAN, sorting networks and multi-heads architecture is interesting.

**Weaknesses:**

However, this paper should be rejected because:
(1) it is built on weak understanding of explainability
(2) there is a weak connection between "explanation disagreement" and the solution
(3) has questionable experiment design and metrics without sufficient justification
(4) it is unrefined

My biggest concern is that the paper uses local explanatory models (i.e. LIME, Integrated Gradients) to generate global (model-level) explanations. In Section 2.3, the authors mention that they have adapted feature attribution methods for local explanations "by averaging feature attributions across all instances to obtain global attributions." These methods were not designed to be used this way. Although SHAP does have functionality to provide model-level feature attribution, it takes an average of the **absolute** attribution across instances.

My impression of the paper's proposed solution, EXAGREE, is that it attempts to address "explanation disagreement" by finding a model that aligns with stakeholder expectations (i.e., based on domain knowledge) through examining its post-hoc explanation. There is one critical assumption here: the post-hoc explanation is faithful to model behavior --- something we cannot take for granted (see e.g. [Adebayo et al. (2019)](https://arxiv.org/abs/1810.03292)). Besides, I don't think the solution addresses the problem of "explanation disagreement", but is rather a model-selection tool using post-hoc explanations. I see that there are two cases of "explanation disagreement" (both of which is mentioned in the paper):
1. Models with similar performance give different explanations (explanation method fixed)
2. Explanation methods provide different explanations for one model (model fixed)
EXAGREE addresses 1 to an extent but not 2 --- the paper does not make this clear. The authors seem to suggest that the "stakeholder centric" approach can address complex disagreements (Section 2.2). But I don't see how it addresses case 2.

Moreover, I am not convinced that "higher agreement between rankings implies greater faithfulness". Bad actors might want explanations that hide the discriminatory nature of their models, hence want features to be ranked a certain way. In fact, several works have highlighted that explainability methods are prone to manipulation:  [Slack et al. (2020)](https://arxiv.org/abs/1911.02508), [Aivodji et al. (2019)](https://proceedings.mlr.press/v97/aivodji19a.html), [Goethals et al. (2023)](https://arxiv.org/abs/2306.13885). Explainability methods are tools to gain insight into a model (to potentially build trust) not project our desired belief upon the model.

As a result, the metrics, which are based on an unsubstantiated assumption that agreement $\implies$ faithfulness, and the empirical results fall short of achieving the goals outlined in the introduction: to identify models "that provide fair faithful and trustworthy explanations."

The experimental design uses the "ground truth" explanation, the coefficients of the LR model, as "stakeholder needs." It is inappropriate to compare this to the explanations of the ANN model. I do not understand why we would want ANN model explanations to agree with LR explanations. Note that this is quite different from what [Agarwal et al. (2022)](https://arxiv.org/abs/2206.11104) did in their experiments. The experiment setup in general is quite confusing.

In line 463, the discussion regards explanation methods as "stakeholders". I question whether it is appropriate to frame it this way as it is difficult to imagine a stakeholder wanting rankings "like LIME".

Furthermore, the paper in its current state does not seem refined. The authors introduce the problem of explanatory multiplicity but do not make an effort to elaborate on how and why it hinders trust in the model (what about the explanation method?). Also, figures 1 and 2 are not helpful in improving the readers' understanding of the EXAGREE process. Figure 1 is especially confusing regarding what it is meant to portray.

**Questions:**

- Section 2.3 describing the metrics should be integrated into the experiment section.
- Is $\psi$ fixed in the optimization problem?
- Line 233-4 "which allows us to transform single attribution values into ranges" -> how? (I know its in the appendix right now, should be in the main body)
	- And why is this an important point to raise?
- Line 469-472: I don't quite understand the significance of this "crucial insight."
- Experiments: what is the $k$ value? I can't seem to find this parameter.
- Neither the discussion nor the figure caption explain what is going on in the figures, what it means and its significance.
- Is there code to reproduce the results?

Nitpicks
- The remarks should be paragraphs. If the authors want to emphasize on a point, the remarks should be more concise.
- 2.3 Evaluation Matrices -> Metrics
- Might want to switch to active voice on some of the sentences

---

> ### Author Response · Authors · 2024-11-14
>
> > My biggest concern is that the paper uses local explanatory models (i.e. LIME, Integrated Gradients) to generate global (model-level) explanations. In Section 2.3, the authors mention that they have adapted feature attribution methods for local explanations "by averaging feature attributions across all instances to obtain global attributions." These methods were not designed to be used this way. Although SHAP does have functionality to provide model-level feature attribution, it takes an average of the  **absolute**  attribution across instances.
>
> Reponse: We appreciate the reviewer's attention to methodological rigor. However, we must respectfully disagree with the concern about aggregating local explanations to obtain global feature importance. This approach has solid theoretical and empirical foundations in the XAI, as demonstrated by following studies:
> -  Section 4 in *Why Should I Trust You?” Explaining the Predictions of Any Classifier*
> - Section 4 in *A Unified Approach to Interpreting Model Predictions*
>  - Section B.2 in *Explaining Explanations: Axiomatic Feature Interactions for Deep Networks*
> - Section 2 in *Interpretability Beyond Feature Attribution: Quantitative Testing with Concept Activation Vectors (TCAV)*
> - *Global Explanations of Neural Network Mapping the Landscape of Predictions*
> - Chapter 3 in book *Interpretable Machine Learning - A Guide for Making Black Box Models Explainable*
> - COMPAS experiment in *Fooling LIME and SHAP: Adversarial Attacks on Post hoc Explanation Methods*
> - Section 3 and 4 in *Fairness via explanation quality: Evaluating disparities in the quality of post hoc explanations*
>
> Moreimportantly, LIME, SHAP, and other local explanation methods are used ONLY as baseline comparisons, following the exact evaluation protocol from the OpenXAI benchmark paper. They are not components of EXAGREE and our contributions stand entirely independent of these comparison methods.
>
> ---
>
> > My impression of the paper's proposed solution, EXAGREE, is that it attempts to address "explanation disagreement" by finding a model that aligns with stakeholder expectations (i.e., based on domain knowledge) through examining its post-hoc explanation... Besides, I don't think the solution addresses the problem of "explanation disagreement", but is rather a model-selection tool using post-hoc explanations. I see that there are two cases of "explanation disagreement" (both of which is mentioned in the paper):
> 	1.  Models with similar performance give different explanations (explanation method fixed)
> 	2.  Explanation methods provide different explanations for one model (model fixed) EXAGREE addresses 1 to an extent but not 2 --- the paper does not make this clear. The authors seem to suggest that the "stakeholder centric" approach can address complex disagreements (Section 2.2). But I don't see how it addresses case 2.
>
> Reponse:  Our intention is not to fully resolve the complex issue of explanation disagreement, but rather to contribute a step toward addressing this challenge. We identify four potential sources of disagreement: model disagreement, method disagreement, stakeholder disagreement, and ground truth disagreement. Given the inherent complexity of these sources, we view the problem as an ongoing research question that warrants further exploration.
>
> We acknowledge that EXAGREE primarily addresses model and stakeholder disagreements (Case 1). In our experiments, we focused on fixing the explanation method to ensure a fair comparison, which means method-based disagreement (Case 2) was not fully explored in this work. We recognize that this is a complex open problem that cannot be resolved with a one-size-fits-all approach.
>
> ---
>
> > Moreover, I am not convinced that "higher agreement between rankings implies greater faithfulness". Bad actors might want explanations that hide the discriminatory nature of their models, hence want features to be ranked a certain way. In fact, several works have highlighted that explainability methods are prone to manipulation:  [Slack et al. (2020)](https://arxiv.org/abs/1911.02508),  [Aivodji et al. (2019)](https://proceedings.mlr.press/v97/aivodji19a.html),  [Goethals et al. (2023)](https://arxiv.org/abs/2306.13885). Explainability methods are tools to gain insight into a model (to potentially build trust) not project our desired belief upon the model.
>
> Reponse: We followed OpenXAI metrics that measures faithfulness through ranking agreement. While we acknowledge that any technique modifying explanations could potentially be misused, as demonstrated in the cited works, the potential for misuse shouldn't prevent responsible research and development, similar to other technological advances in AI.
> EXAGREE's optimization within the Rashomon set offers a distinct perspective: unlike methods that produce single rankings, it helps users understand the range of possible feature attributions (attribution sets), enabling more informed decisions with confidence.

---

> > ### Author Response · Authors · 2024-11-14
> >
> > > As a result, the metrics, which are based on an unsubstantiated assumption that agreement  ⟹  faithfulness, and the empirical results fall short of achieving the goals outlined in the introduction: to identify models "that provide fair faithful and trustworthy explanations."
> >
> > Response: Our approach builds upon established OpenXAI metrics.
> > Consider a scenario where we have:
> >
> > -   Known ground truth feature importance ranking (from LR or domain knowledge): A > B > C
> > -   ML model (ANN) producing importance ranking (e.g., via SHAP): C > B > A
> >
> > As a user, he knows A is more important but only has access to ANN. The disagreement between these two rankings directly reflects a lack of faithfulness.
> >
> > ---
> >
> > > The experimental design uses the "ground truth" explanation, the coefficients of the LR model, as "stakeholder needs." It is inappropriate to compare this to the explanations of the ANN model. I do not understand why we would want ANN model explanations to agree with LR explanations. Note that this is quite different from what  [Agarwal et al. (2022)](https://arxiv.org/abs/2206.11104)  did in their experiments. The experiment setup in general is quite confusing.
> >
> > Response: Regarding the comparison between LR coefficients and ANN explanations, we would like to clarify our experimental setup and its practical motivation.
> >
> > Imagine the user is given the above ANN that produces explanations (C > B > A) inconsistent with his domain knowledge (e.g., claiming feature C is most important when he knows A should be). In such cases, the user would naturally want to find another ML model whose explanations align with the known feature importance relationships. This is precisely what our experimental setup evaluates.
> >
> > ---
> >
> > > In line 463, the discussion regards explanation methods as "stakeholders". I question whether it is appropriate to frame it this way as it is difficult to imagine a stakeholder wanting rankings "like LIME".
> >
> > This is indeed an assumption we made to systematically evaluate different perspectives on feature importance. Referring to the above example, domain knowledge of another user might result in a ranking B > A > C, which we consider as "Like LIME" for experimental purposes only.
> >
> > ---
> >
> > >Furthermore, the paper in its current state does not seem refined. The authors introduce the problem of explanatory multiplicity but do not make an effort to elaborate on how and why it hinders trust in the model (what about the explanation method?). Also, figures 1 and 2 are not helpful...
> >
> > The paper builds upon the general understanding of the explanation disagreement problem. This issue becomes problematic when differences in explanations are unrecognized and explanations are trusted blindly. For example, in the criminal justice system, an ML model could be manipulated to base decisions on race, with severe consequences for societal fairness and justice.
> >
> > The curved arrows, lightbulb, question mark, and check mark were intended as visual aids to represent explanation disagreement problem and potential solver. We will consider either removing Figure 1 or remaking it, and we will revise Figure 2 to better illustrate the core framework.
> >
> > ---
> >
> > **Questions:**
> >
> > > Section 2.3 describing the metrics should be integrated into the experiment section.
> >
> > The metrics section serves dual purposes 1. general metrics for explanation disagreement problem and 2. being used in our experiments e.g., fixed explanation method. We will restructure to clearly this point.
> >
> > >Is $\varphi$ fixed in the optimization problem?
> >
> > Yes, the explanation method is fixed to ensure a fair comparison.
> >
> > >Line 233-4 "which allows us to transform single attribution values into ranges" -> how? (I know its in the appendix right now, should be in the main body)
> >
> > The Rashomon set contains models with different feature attributions. For each feature, we get an attribution range across all models, rather than a single value. This is crucial as it defines the feasible space for our optimization framework while maintaining model performance.
> >
> > >Line 469-472: I don't quite understand the significance of this "crucial insight."
> >
> > The observation shows why a single model/explanation can't satisfy conflicting but valid stakeholder perspectives (e.g., clinical vs medical researcher). This example illustrates stakeholder disagreement and reinforces the motivation behind our approach.
> >
> > >Experiments: what is the k  value? I can't seem to find this parameter.
> >
> > We apologize for this oversight. $k$ represents the top percentage of features considered. For example, with $k$=0.25 in a list of 12 features, we consider the top 3 most important features
> >
> > >Neither the discussion nor the figure caption explains what is going on in the figures, what it means and its significance.
> >
> > We will enhance both the discussion and figure captions to better explain their significance and interpretation.
> >
> > >Is there code to reproduce the results?
> >
> > Yes, it will be released soon.

---

> > > ### Comment · Reviewer_ryTs · 2024-11-24
> > >
> > > I echo the concerns raised by Reviewer **9c6d**: the authors seem to have a critical misunderstanding of the relevant literature. I will point out a few:
> > >
> > > 1. The aggregation in Section 4 of _Why Should I Trust You? Explaining the Predictions of Any Classifier_ is **NOT** what the authors have performed in the paper. [Ribeiro et al. 2016](https://arxiv.org/pdf/1602.04938) aims to explain the entire model by explaining a subset of instances.
> > > 2. Section 4 of _A Unified Approach to Interpreting Model Predictions_ doesn't mention aggregation to provide a global explanation of the model.
> > > 3. The COMPAS experiment in _Fooling LIME and SHAP: Adversarial Attacks on Post hoc Explanation Methods_ does **NOT** take the aggregate. The plots show "% of data points for which each feature (color coded) shows up in the top 3 (according to LIME and SHAP’s ranking of feature importance)" (taken directly from the caption of Figure 2 in [Slack et al. 2019](https://arxiv.org/pdf/1911.02508)).
> > >
> > > > As a user, he knows A is more important but only has access to ANN. The disagreement between these two rankings directly reflects a lack of faithfulness.
> > >
> > > Faithfulness of an explanation usually refers to how accurate the explanation is to **model** behavior, not "ground truth." This is because the goal of explainability is to understand model behavior, which may not necessarily align with the ground truth (and that's fine). The goal of machine learning is to build well performing models (see [Breiman 2001](https://projecteuclid.org/journals/statistical-science/volume-16/issue-3/Statistical-Modeling--The-Two-Cultures-with-comments-and-a/10.1214/ss/1009213726.full)).
> > >
> > > Also, to my knowledge, the idea of the **Rashomon effect** was introduced in this paper. In any case, I recommend the authors read this work if they haven't already.
> > >
> > > > Imagine the user is given the above ANN that produces explanations (C > B > A) inconsistent with his domain knowledge (e.g., claiming feature C is most important when he knows A should be). In such cases, the user would naturally want to find another ML model whose explanations align with the known feature importance relationships. This is precisely what our experimental setup evaluates.
> > >
> > > This is right in line with my point that EXAGREE "is rather a model-selection tool using post-hoc explanations."

---

> > > > ### Author Response · Authors · 2024-11-25
> > > >
> > > > We appreciate the reviewer’s feedback. Our intention was to demonstrate that aggregating local explanations to derive global insights is a recognized practice in the field. While the cited works do not explicitly aggregate explanations in the same way, they contextualize related methodologies (e.g., [Okeson et al., Summarize with Caution, 2022](https://www.microsoft.com/en-us/research/uploads/prod/2022/01/Summarize-with-Caution_Comparing-Global-Feature.pdf)).
> > > >
> > > > We acknowledge the distinction highlighted regarding faithfulness, which refers to alignment with model behavior rather than ground truth, and will ensure this is clarified. Additionally, we will refine our evaluation by exploring metrics such as normalized LIME or computing instance-level metrics before aggregation.
> > > >
> > > > We thank the reviewer for these insights, which will help improve the clarity and rigor of our work.

---

### Official Review · Reviewer_3uFb · 2024-11-08

**Soundness:** 2
**Presentation:** 1
**Contribution:** 2
**Rating:** 3
**Confidence:** 3

**Summary:**

The paper titled "EXAGREE: Towards Explanation Agreement in Explainable Machine Learning" addresses the challenge of explanation disagreement in machine learning. Explanation disagreement, where model explanations diverge based on methods, models, or stakeholder expectations, hampers trust and transparency in high-stakes environments. The authors propose a framework called EXplanation AGREEment (EXAGREE) that utilizes a Rashomon set—multiple models with similarly good predictive performance—to align explanations with diverse stakeholder expectations. By optimizing within this set, EXAGREE identifies Stakeholder-Aligned Explanation Models (SAEMs) that reduce disagreement while preserving predictive accuracy.

The authors formalize four types of explanation disagreement: stakeholder, model, explanation method, and ground truth disagreements. EXAGREE addresses these by introducing a two-stage process: Rashomon set sampling, followed by SAEM identification. Empirical analyses demonstrate that EXAGREE reduces explanation disagreements and improves fairness across datasets, positioning it as a potentially valuable tool for practitioners aiming to enhance trust and fairness in machine learning applications​.

**Strengths:**

- The paper addresses a critical challenge faced by researchers and practitioners: how to proceed when even explainable AI tools disagree on feature importance. Additionally, it incorporates model and stakeholder rankings, making the approach quite comprehensive.
- The paper tackles its proposed problem by integrating methodologies from several different areas, including the XAI literature as well as general AI methods for optimization challenges.
- The insight to make the process end-to-end differentiable is both creative and practically useful.
In the appendix, the authors demonstrate the impact of the choice of $\epsilon$ on the Rashomon set, which serves as a valuable methodological sensitivity analysis.
- The empirical results test their methods across a variety of settings: 6 OpenXAI datasets, both synthetic and empirical; and 2 pre-trained models (logistic regression and artificial neural networks).

**Weaknesses:**

The paper integrates several methods and techniques to address the explainability disagreement issue; however, it can feel somewhat dry and lacks the depth and technical details that would enable readers to fully appreciate the contributions and identify strengths and weaknesses. Technical jargon used to describe the methodology needs precise definitions, mathematical arguments should be clearly defined and explained, and additional background information would offer useful entry points for readers. Most of the following points align with this suggestion. This lack of precise definitions also contributes to my limited confidence in the recommendations, as it made it challenging to fully assess the work's potential impact.
- The loss functions for $L_{sparsity}$ and $L_{diversity}$ are not defined clearly in the paper or the appendix.
- Precise mathematical definitions of what a mask is and how it is derived are essential for readers to understand the methodology in depth, as this concept is central to the approach.
- Consider adding a sentence or footnote to define core terms in your algorithm, as these abstract concepts can vary in meaning:
    - "attribution set", "model representations", "model characterizations", "end-to-end optimization".
    - For instance, in the sentence "Training a Differentiable Mask-based Model to Attribution Network (DMAN) that maps feature attributions from model characterizations for use in the next stage," it would be helpful to clarify precisely what “model characterizations” entail.
    - Also, in Equation 4, where $f_{DMAN}^*$ is defined as the optimal surrogate model in the Rashomon set that describes feature attributions, it appears to be the loss between $f_{DMAN}$ and a set comprising ${\text{masks}, \text{attributions}}$. Minimizing the output to such abstractly defined elements would benefit from more clarity.
- It would be helpful to include insights or references for the result in row 196. Why is faithfulness proportional to agreement? Is this a theoretical result, an empirical finding from the paper, or something else?
- Figure 1 provides few entry points for readers and doesn’t seem to aid in understanding at its current placement. Consider either removing it or adding more descriptive captions to clarify each step (similar to Figure 2, which includes more context). Suggestions include captions for the rankings, lightbulb, etc. Additionally, why are stakeholders grouped together in the first half but not in the second half?
- Figure 2 is clear, but it seems to appear too early in the paper. Moving it to the end of Section 3 might make it more helpful, as readers would have more context to interpret it.
- Is the fairness improvement an explicit objective of the EXAGREE model? If so, please explain the rationale and mechanism. If it’s an outcome of the empirical analysis, please clarify this in the paper, as empirical results may not generalize across all applications.
- In row 276, in the explanation of the "Ranking Supervision and Correlation Metric," it would be beneficial to provide more context and motivation for this metric and how it fits in the big picture of your methodology before defining it.

**Questions:**

Specific suggestions (please feel free to disregard these if I’ve misunderstood something):
- Typo in row 480: “Rashomon” is misspelled.
- Possible typo or confusion in row 340: “We utilized two pre-trained models,” but immediately afterward, three model types are mentioned.
- Readability suggestion: Consider defining $f_{diffsort}$ (f_{diffsort}) as $f_{\text{diffsort}}$ (f_{\text{diffsort}}) for readability and saving space in equations?

---

> ### Author Response · Authors · 2024-11-13
>
> We thank the reviewer for the feedback and appreciate the recognition of reducing explanation disagreement through our framework. While the reviewer raises valid concerns regarding technical details, we prioritized problem formalization and overall framework development within the page constraints. We address the main points raised:
> >The loss functions for $L_{sparsity}$ and $L_{diversity}$ are not defined clearly in the paper or the appendix.
>
> Response: $L_{sparsity}$ controls the distribution of values across different masks and $L_{diversity}$ maximizes the variance within each mask, ensuring they focus on distinct feature subsets. We will refine their definitions mathematically.
>
> > Precise mathematical definitions of what a mask is and how it is derived are essential for readers to understand the methodology in depth, as this concept is central to the approach.
>
> Response: Given space constraints, we focused more on novel contributions over technical details available in prior work. We recognize the need for clarity on the mask concept and will include precise mathematical definitions in the Appendix.
>
> Simply put, a mask characterizes/represents a model’s unique architecture within the Rashomon set by capturing distinct feature attribution patterns. This abstraction layer allows EXAGREE to compare and optimize models in the Rashomon set against stakeholder rankings and expectations. Such a mask can be obtained by adding additional layers to the optimal (reference) model, while ensuring that the resulting model still meets the loss condition within the Rashomon set. Equation (1) $L(M(X), y)$ can be adjusted to $L(M^{*} \circ m(X), y)$ for understand, where $m$ represents masks.
>
> >Consider adding a sentence or footnote to define core terms in your algorithm, ... as these abstract concepts can vary in meaning.
>
> Response: We agree with the reviewer's suggestion and include a brief discussion here.
> - **Attribution Set**: In our framework, each model generates a ranked list of feature attributions. The combined feature attributions from all models within the Rashomon set form what we refer to as an attribution set.
> - **Model Representations** &**Model Characterizations**: In EXAGREE, masks serve as representations and characterizations for models within the Rashomon set. Each mask corresponds to a specific model, producing associated feature attributions. This structure allows for consistent model comparisons and  end-to-end optimization.
> - **End-to-End Optimization**: A training approach where all components of the framework are optimized simultaneously to minimize a specified loss function.
> - In Equation 4, the surrogate model is designed to learn the non-linear relationship between feature attributions and their associated masks in the Rashomon set, enabling further optimization.
>
> > It would be helpful to include insights or references for the result in row 196. Why is faithfulness proportional to agreement? Is this a theoretical result, an empirical finding from the paper, or something else?
>
> Response: The relationship follows OpenXAI's faithfulness metric definition, where faithfulness is measured by the agreement between rankings. Therefore, higher agreement with ground truth rankings naturally leads to higher faithfulness scores. We will add more discussion in the revised paper.
>
> > Figure 1.... Additionally, why are stakeholders grouped together in the first half but not in the second half?
>
> Response: We will revise the figure and captions. We aim to illustrate the disagreement problem through grouped stakeholders with conflicting interests, while in the second half, individual needs are highlighted to represent our objective of satisfying diverse stakeholder requirements.
>
> > Figure 2 is clear, but it seems to appear too early in the paper...
>
> Response: We will relocate this figure to Section 3.
>
> >  Is the fairness improvement an explicit objective of the EXAGREE model?...
>
> Response: This is an empirical outcome rather than an explicit objective for the current framework. We will clarify this and discuss its implications.
>
> >In row 276, in the explanation of the "Ranking Supervision and Correlation Metric," it would be beneficial to provide more context and motivation for this metric.
>
> Response: Our choice of a ranking correlation metric is motivated by its connection to OpenXAI's faithfulness metrics, where faithfulness is measured through ranking agreement. We specifically selected Spearman's rank correlation because it provides a differentiable measurement and is well-established in the literature. We will revise Section 3.2 to include more motivation and discussion.
>
> >Specific suggestions
>
> - Rashomon is corrected.
> - Clarified the use of two pre-trained models plus one interpretable model.
> - Refined the notation for $f_{\text{diffsort}}$.
> ---
> We hope these clarifications are helpful and welcome any further discussion to strengthen the paper.

---

> > ### Comment · Reviewer_3uFb · 2024-11-26
> >
> > Thanks for the clarifications. I still believe that the lack of these precise definitions in the paper (which could be shrunk in some parts to fit these short definitions) hampers technical understanding and does not match the rigor of an ICLR paper. The author has to be precise about its central objects, so the general audience is able to understand in depth the strengths and weaknesses of the model. Your idea seems to be great, you just need to work a bit more on the formal aspects of the paper presentation.

---

### Author Response · Authors · 2024-11-13

Dear Reviewers,

We sincerely thank you for the time and effort invested in reviewing our work. Although we were surprised by the range of scores assigned, we would like to clarify the broader context and significance of our contributions, which we believe were not fully appreciated.

1. **Context and Motivation**: Explanation disagreement arises when conflicting explanation outputs are produced, potentially leading to significant trust risks. We study the problem as misleading or inconsistent explanations can compromise trust and result in the deployment of biased or unsafe models. For example, in the criminal justice system, an ML model could be manipulated to base decisions on race, with severe consequences for societal fairness and justice [1].

2. **Problem Setup**: Our work formalizes the ranking-based explanation disagreement problem for discussion, and another concise example in the healthcare context is provided below.

3. **Our Approach**: To mitigate this problem, we proposed that there exists a model that provides better explanations for specific stakeholders within a set of well-performing models (a Rashomon set). We achieved this objective by EXAGREE framework. We stress that this is an open problem, and our goal is to advance understanding rather than claim a definitive solution.

4. **Evaluation**: We used OpenXAI, one of the latest benchmarks for faithfulness based on ranking agreements, including FA, RA, SA, SRA, PRA. We demonstrated both the existence of disagreement and visualized agreement improvements.



We hope this overview provides a clearer understanding of our work’s intent and structure. While our study does not exhaustively address every aspect of explanation disagreement, we believe it provides valuable empirical insights and a foundation for further research. We kindly request reviewers to consider our work as a step toward exploring a new direction of trustworthy XAI.

---
**Example**: Breast Cancer Prediction.

Consider a hospital deploying ML models to assist in breast cancer diagnosis, with multiple stakeholders involved.

*Models*: Decision Tree, Neural Network, XGBoost; *Stakeholders*: Model developer, Medical researcher, Regulator, Clinician, Patient

1. Model disagreement: all three models are trained with similar and promising performance, same post-hoc explanations of these models are different. For instance, the Neural Network might emphasize texture patterns, while the Decision Tree focuses more on cell size measurements.



2. Method disagreement: explaining the Decision Tree with different post-hoc explanations (SHAP, LIME) provides different explanations.



3. Ground Truth disagreement: The Decision Tree's intrinsic explanation (decision paths) might indicate age and tumor size as key factors, while post-hoc methods suggest different feature importances.



4. Stakeholder disagreement: Different stakeholders care about different features
	 - Model Developers focus on which features give the highest predictive accuracy

	 - Medical Researchers care about features aligned with biological mechanisms

	 - Regulators need to verify which demographic features influence predictions

	 - Clinicians want to know which measurable clinical features drive decisions

	 - Patients need to understand which personal health indicators affect their diagnosis

References:

[1] Slack, D., Hilgard, S., Jia, E., Singh, S., & Lakkaraju, H. (2020, February). Fooling lime and shap: Adversarial attacks on post hoc explanation methods. In Proceedings of the AAAI/ACM Conference on AI, Ethics, and Society (pp. 180-186).

[2] Imrie, F., Davis, R., & van der Schaar, M. (2023). Multiple stakeholders drive diverse interpretability requirements for machine learning in healthcare. Nature Machine Intelligence, 5(8), 824-829.

---

> ### Comment · Reviewer_9c6d · 2024-11-13
>
> I would be grateful if the authors could answer at least some of the specific concerns raised in the reviews. The citation of [1] continues to be mis-placed, and as per my understanding strengthens the critique of the paper, rather than supporting the current way of framing the contributions.

---

### Meta-Review · Area_Chair_a2v9 · 2024-12-20

**Metareview:**

All reviewers appreciated the high-level goal of the paper - a form of model selection that maintains strong performance while adhering to an explainability constraint that aims to balance potentially-competing stakeholders' wants.  That said, all reviewers felt the method as presented was not detailed/correctly placed in the literature enough, especially given the complexity of the SAEM pipeline.  This AC is empathetic toward the difficult of presenting complex end-to-end pipelines, and lauds the authors for doing so, but agrees with all reviewers that the work needs another iteration -- as presented, the pipeline itself is ambiguously presented and defined.

**Additional Comments On Reviewer Discussion:**

We appreciate the authors' and the reviewers' participation during the discussion phase.  There was back and forth about details of the SAEM pipeline as well as placement in the literature, but reviewers remained unconvinced that this is ready in its current form.

---

### Decision · Program_Chairs · 2025-01-22

Reject